# The Information Geometry of Softmax: Probing and Steering

Kiho Park [1]   Todd Nief [1]   Yo Joong Choe [2]   Victor Veitch [1]

## Abstract

This paper concerns the question of how AI systems encode semantic structure into the geometric structure of their representation spaces. The motivating observation is that the natural geometry of these representation spaces should reflect the way models use representations to produce behavior. We focus on the important special case of representations that define softmax distributions. In this case, we argue that the natural geometry is information geometry. Our focus is on the role of information geometry on semantic encoding and the linear representation hypothesis. As an illustrative application, we develop *dual steering*, a method for robustly steering representations to exhibit a particular concept using linear probes. We prove that dual steering optimally modifies the target concept while minimizing changes to off-target concepts. Empirically, we find that dual steering enhances the controllability and stability of concept manipulation. Code is available at github.com/KihoPark/dual-steering.

## 1. Introduction

Understanding and manipulating the internal representations of AI models is central for building trustworthy and controllable AI systems. Many approaches build on the *linear representation hypothesis*—the idea that high-level concepts (e.g., sentiment, truthfulness, or gender) correspond to specific directions in the vector space containing the model's representations (Mikolov et al., 2013b; Elhage et al., 2022; Park et al., 2024). Researchers have used this idea to identify and manipulate concepts across various architectures (Nanda et al., 2023; Li et al., 2023; Turner et al., 2023; Zou et al., 2023; Gurnee & Tegmark, 2024). However, the results are somewhat mixed. Although there is clearly structure in the representation spaces, these methods are often brittle, and have usually not been competitive with more direct

---

[1]University of Chicago [2]INSEAD.

*Proceedings of the 43rd International Conference on Machine Learning*, Seoul, South Korea. PMLR 306, 2026. Copyright 2026 by the author(s).

fine-tuning approaches (Hase et al., 2023; Makelov et al., 2024; Sharkey et al., 2025; Wang & Veitch, 2025). This suggests that we do not yet have a full enough understanding of the 'linear representation' structure to build robust, generalizable methods.

One gap in our understanding is that linear representation methods are frequently built on the (implicit) assumption that the representation space has a flat (or even Euclidean) geometry, but there is little reason to expect this assumption to hold. Instead, we would like methods based on the 'intrinsic' structure of the representation space. To that end, we need a notion of geometry that aligns with the way the model actually uses its representations to produce behavior—e.g., a geometry where two representations are 'close' if they produce similar outputs. The purpose of this paper is to operationalize this idea in the particular case of softmax based models, and to explain the practical implications of the resulting geometry for interpretability methods.

Our focus here is on representation vectors $\lambda \in \Lambda \simeq \mathbb{R}^d$ that define probability distributions via the softmax transform. That is, for any set of candidate items $\mathcal{Y}$, the model assigns $\{\gamma_1, \gamma_2, \ldots, \gamma_{|\mathcal{Y}|}\} \subset \Gamma \simeq \mathbb{R}^d$ as vector representations of each item, and defines the softmax probability distribution:

$$\mathbb{P}(\gamma = \gamma_y \mid \lambda) = \exp\big(\lambda^\top \gamma_y - A(\lambda)\big), \qquad (1)$$

where $A(\lambda) := \log \sum_{y'} \exp(\lambda^\top \gamma_{y'})$ is the log-normalizer. This pattern shows up in many AI architectures, including in the attention mechanism of transformers (Vaswani et al., 2017), the next-token selection of large language models (LLMs) (Brown et al., 2020), and contrastive models like CLIP (Radford et al., 2021). Our starting point is the observation that the notion of closeness of two representation vectors $\lambda, \lambda'$ should reflect the closeness *of the induced probability distributions*. Information geometry provides a powerful framework for formalizing and studying the innate geometry of parameters of probability distributions (Amari & Nagaoka, 2000; Banerjee et al., 2005; Amari, 2016). The main aim of this paper is to understand how the linear representation hypothesis—and the encoding of high-level semantics in representation space—interacts with the natural information geometry of the representation space.

To that end:

1. We identify the natural geometry as a Bregman (dually

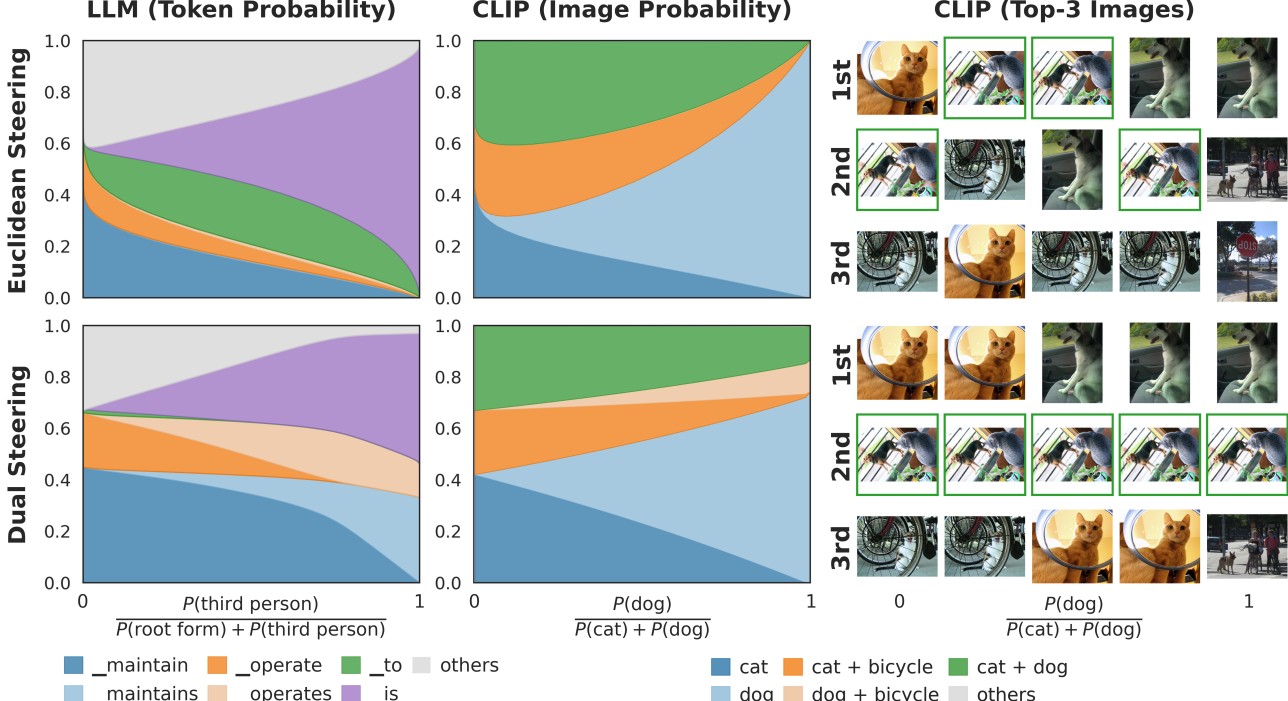

*Figure 1.* **Dual steering (bottom) effectively modifies the target concept (e.g., `verb` ⇒ `third-person` or `cat` ⇒ `dog`) while preserving off-target distributions (e.g., $P(\text{"maintain"}) + P(\text{"maintains"})$ or $P(\text{"cat + bicycle"}) + P(\text{"dog + bicycle"})$), whereas Euclidean steering (top) fails to maintain off-target distributions despite reaching the target concept probability. Left:** Token probability changes in Gemma-3-4B when steering the context "Author gives an insight into what it costs US taxpayers to build and" using a linear probe for `verb` ⇒ `third-person`. Euclidean steering leaks significant mass to off-target tokens (e.g., "to") during intermediate steps, whereas dual steering directly shifts probability from base tokens (e.g., "maintain", "operate") to target tokens (e.g., "maintains", "operates"). **Center & Right:** Steering MetaCLIP-2 on the context "a photo of one cat" for the concept `cat` ⇒ `dog`. Dual steering transfers probability from base images (e.g., "cat", "cat + bicycle") directly to targets (e.g., "dog", "dog + bicycle"). In contrast, Euclidean steering unintentionally promotes the off-target "cat + dog" image (green frame in the right column), which becomes the Top-1 result during intermediate steps. In the probability plots, Top-$K$ tokens (LLM) or images (CLIP) are shown explicitly, with the remainder grouped as "others."

flat) geometry. This induces a rich duality structure that will play a critical role in understanding the semantic structure of the representation space.

2. We then study the question of how to interpolate between two representation vectors. In short: there are natural distinct primal and dual interpolations that yield distinct semantics. In particular, this dual interpolation structure shows that a flat geometry cannot suffice to capture the semantic structure of the representation space.

3. We then show how information geometry interacts with probing and steering representation vectors. This leads us to "dual steering", a new method for robustly manipulating representations. We prove that this method modifies the target concept while minimizing unintended changes to off-target concepts.

4. Finally, we test dual steering using open-source models, including Gemma-3-4B (Kamath et al., 2025) and

MetaClip-2 (Chuang et al., 2025), showing improved controllability and stability relative to standard Euclidean steering approaches; see Figure 1.

The high-level observation here is that the non-Euclidean structure of the intrinsic information geometry of representation vectors is critical for connecting geometry and semantic tasks such as steering. Although the results here only apply directly to softmax-based models, the high-level idea is widely applicable. Then, in part, we hope that this work can serve as a template for exploiting geometry to improve the robustness of interpretability and control methods more generally.

## 2. Duality and Interpolation

We begin by introducing the information geometric structure and studying the (comparatively) simple problem of interpolation between two representation vectors.

## 2.1. Bregman Duality Induced by Softmax

Our starting observation is that the Kullback-Leibler (KL) divergence between softmax distributions (1) can be expressed as

$$D_{\mathrm{KL}}\left(P_\lambda \parallel P_{\lambda'}\right) = A(\lambda') - A(\lambda) - \nabla A(\lambda)^\top (\lambda' - \lambda),$$

where $P_\lambda := \mathbb{P}(\gamma = \cdot \mid \lambda)$. This relation can be readily checked by direct computation. The important observation is that the right-hand side is the *Bregman divergence* induced by the convex function $A$. That is, the representation geometry induced by the KL divergence is a Bregman (or dually flat) geometry.

For our purposes, the key aspect of the Bregman geometry will be its rich duality structure. For a context embedding $\lambda$, the *dual map* is defined by the gradient of $A$:

$$\phi(\lambda) := \nabla A(\lambda) = \mathbb{E}[\gamma \mid \lambda].$$

When $A$ is strictly convex, we also have an *inverse map* as

$$\lambda(\phi) := \nabla A^*(\phi), \quad \text{so that} \quad \nabla A^*(\phi(\lambda)) = \lambda,$$

where $A^*$ is a convex conjugate of $A$ over the image of $\nabla A$:

$$A^*(\phi) := \sup_\lambda \left\{ \lambda^\top \phi - A(\lambda) \right\}, \quad \phi \in \Phi := \mathrm{Image}(\nabla A).$$

Together, these mappings provide a bijection between the primal space $\Lambda$ and the dual space $\Phi$. A *primal coordinate* $\lambda$ and its *dual coordinate* $\phi(\lambda)$ are different parameterizations of the same probability distribution $P_\lambda$ (or $P_{\phi(\lambda)}$).

## 2.2. Interpolation in Primal and Dual Spaces

Our ultimate goal is to connect this geometric framework to the semantic structure of the representation space. To that end, we begin by studying what it means to interpolate between two points in the representation space.

In the context of Bregman geometry, there are two natural ways to interpolate between two points: in the primal space and in the dual space. For two given context embeddings $\lambda_0$ and $\lambda_1$, the straight line between them in the primal space is called an *e-geodesic*:

$$\lambda_t = (1-t)\lambda_0 + t\lambda_1, \quad t \in [0,1],$$

which is a primal interpolation. On the other hand, the straight line between the dual coordinates $\phi(\lambda_0)$ and $\phi(\lambda_1)$ in the dual space is called an *m-geodesic*:[1]

$$\phi_t = (1-t)\phi(\lambda_0) + t\phi(\lambda_1), \quad t \in [0,1],$$

---

[1] These "geodesics" are not the shortest path with respect to a Riemannian metric, but rather they are defined by specific affine connections (Amari, 2016).

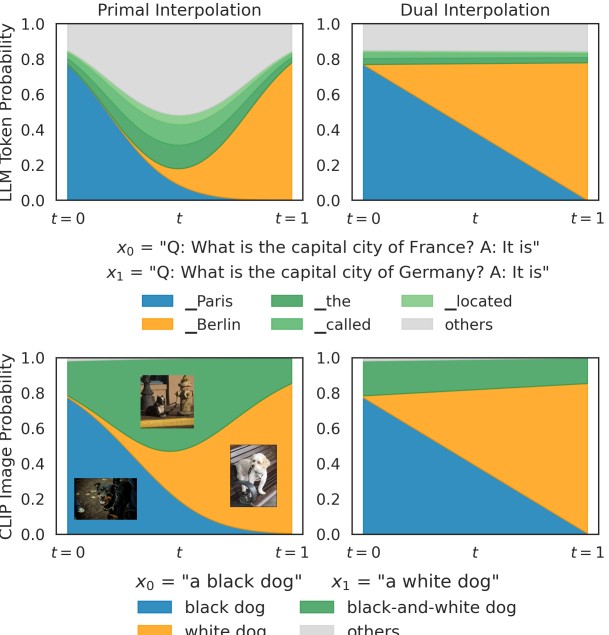

*Figure 2.* **Primal interpolation emphasizes the shared structure (intersection) of distributions, whereas dual interpolation results in a linear mixture.** We visualize output probability changes along interpolation paths between two context embeddings $\lambda(x_0)$ and $\lambda(x_1)$. The **dual interpolation** (right, $m$-geodesic: $\phi_t = (1-t)\phi(\lambda_0) + t\phi(\lambda_1)$) corresponds to a weighted average of the endpoint distributions. In contrast, near the midpoint, the **primal interpolation** (left, $e$-geodesic: $\lambda_t = (1-t)\lambda_0 + t\lambda_1$) up-weights shared components (e.g., "the", "called" in LLM or "black-and-white dog" in CLIP), while suppressing endpoint-specific outputs (e.g., "Paris" vs. "Berlin," or "black dog" vs. "white dog"). Top-$K$ tokens (LLM) or images (CLIP) are shown explicitly, with the remainder grouped as "others."

which is a dual interpolation. The $e$- and $m$-geodesics represent two distinct interpolation paths on the statistical manifold; the primal interpolation will generally be non-linear in the dual coordinate system, and vice versa.

Their behaviors are closely related to the minimization of KL divergences, as summarized in the following proposition:

**Proposition 2.1** (Interpolation as KL Minimization). *The primal interpolation* $\lambda_t$ *minimizes a weighted sum of* ***reverse KL divergences***:

$$\lambda_t \in \operatorname*{argmin}_{\lambda \in \Lambda}(1-t)D_{\mathrm{KL}}\left(P_\lambda \parallel P_{\lambda_0}\right) + tD_{\mathrm{KL}}\left(P_\lambda \parallel P_{\lambda_1}\right),$$

*whereas the dual interpolation* $\phi_t$ *minimizes a weighted sum of* ***forward KL divergences***:

$$\phi_t \in \operatorname*{argmin}_{\phi \in \Phi}(1-t)D_{\mathrm{KL}}\left(P_{\lambda_0} \parallel P_\phi\right) + tD_{\mathrm{KL}}\left(P_{\lambda_1} \parallel P_\phi\right).$$

All proofs are provided in Section A.

The difference between these minimization objectives leads to fundamentally distinct behaviors during interpolation. Consider approximating a target distribution $P$ with a distribution $Q$ by minimizing either the reverse KL, $D_{\mathrm{KL}}(Q \parallel P)$, or the forward KL, $D_{\mathrm{KL}}(P \parallel Q)$. If $Q$ assigns high probability to events that are unlikely under $P$, the reverse KL $D_{\mathrm{KL}}(Q \parallel P)$ becomes very large. Conversely, if $Q$ assigns low probability to events that are likely under $P$, the forward KL $D_{\mathrm{KL}}(P \parallel Q)$ becomes very large. Consequently, the reverse KL minimizer—and thus, the primal interpolation—tends to capture the intersection of high-probability regions, behaving like an AND operator. In contrast, the forward KL minimizer—and thus, the dual interpolation—tends to take the union of high-probability regions, behaving like an OR operator.

**Interpolation Results on LLMs and CLIP Models** Figure 2 illustrates the distinction between primal and dual interpolation in the context of LLMs and CLIP models. At the midpoint of the primal interpolation, we observe that the Top-$K$ tokens (or images) with high probability represent an intersection of the possible outputs from both contexts. Tokens exclusive to only one context have their probabilities significantly suppressed, while those consistent with both contexts are amplified. For example, when $x_0$ = "a black dog" and $x_1$ = "a white dog", the predicted probability for an image of a black-and-white dog is substantially higher at the primal midpoint than at either endpoint. Conversely, in the dual interpolation, the probability mass is more evenly distributed across the union of the possible outputs from both contexts. This indicates that the dual interpolation preserves the semantic features of both contexts simultaneously rather than filtering for their overlap. Formally, this dual interpolation corresponds to a linear mixture of the two distributions.

## 3. Dual Steering with a Linear Probe

The interpolation results show that the duality structure plays a crucial role in capturing the semantic structure of the representation space. We now turn to the question of how the information geometry interacts with the linear representation hypothesis and, in particular, steering representations to exhibit particular concepts.

We will focus on contrastive binary concepts such as `male ⇒ female` or `dog ⇒ cat`. Taking `dog ⇒ cat` as an example, $W = 0$ corresponds to the base concept 'dog' and $W = 1$ corresponds to the target concept 'cat'. We will assume that we have identified a *linear probe* $\beta_W$ that captures the concept. Specifically,

$$P(W = 1 \mid \lambda) = \sigma(\beta_W^\top \lambda + b_W) \quad \forall \lambda \in \Lambda, \qquad (2)$$

where $b_W$ is a concept-specific offset. This relation is basically the defining property of logistic regression, matching

a standard approach to designing linear probes (Alain & Bengio, 2016; Li et al., 2023). In general, it is unclear what makes an ideal probe or how best to identify one. For our analysis, we will simply assume that such a probe has been identified in some manner. The question we address here is: given such a probe, how should we manipulate a given representation to change the desired concept?

The standard approach is to simply add the probe vector directly to the representation:

$$\lambda_t := \lambda_0 + t\beta_W, \quad t > 0. \qquad (3)$$

We will refer to this as *Euclidean steering*. It is clear that for sufficiently large $t$, we will have $\beta_W^\top \lambda_t \gg 0$, and thus $P(W = 1 \mid \lambda_t) \approx 1$. Accordingly, if the probe is well-aligned with the concept, this method should successfully steer the representation to express the desired concept. However, this form of steering may also induce undesirable off-target effects, changing other concepts in unintended ways.

Indeed, there is a basic type mismatch in (3). The probe vector $\beta_W$ is an element of the dual space (it's a linear operator on $\Lambda$), but we are naively adding it to an element of the primal space. This is only valid in the special case where the primal and dual spaces coincide, i.e., when the geometry is Euclidean. This observation motivates us to introduce *dual steering*, which adds the probe vector in the dual space:

$$\phi(\lambda_t) := \phi(\lambda_0) + t\beta_W, \quad t > 0.$$

As we now show, this dual steering approach is robust in the sense that it minimally perturbs off-target behavior.

### 3.1. Robustness of Dual Steering

The goal of steering is to modify the on-target concept while minimizing changes to off-target concepts. With a probe in hand, we can formalize modifying the on-target concept as moving $\lambda_0$ to $\hat{\lambda}_c$ such that $\beta_W^\top \hat{\lambda}_c = c$ for some target $c$. Then, it is natural to view steering as solving the constrained optimization problem:

$$\hat{\lambda}_c = \underset{\lambda : \beta_W^\top \lambda = c}{\arg\min} D(\lambda, \lambda_0),$$

where $D$ is some notion of distance on the representation space. If $D$ is the Euclidean distance, the solution corresponds to Euclidean (standard) steering (3). This formalizes "off target" movement as Euclidean distance. However, there is no reason that this should be the correct notion of distance. We now show that dual steering arises from minimizing a principled notion of off-target change.

We begin by considering what it means to preserve "off-target" concepts. Consider a context "He is my" that predicts

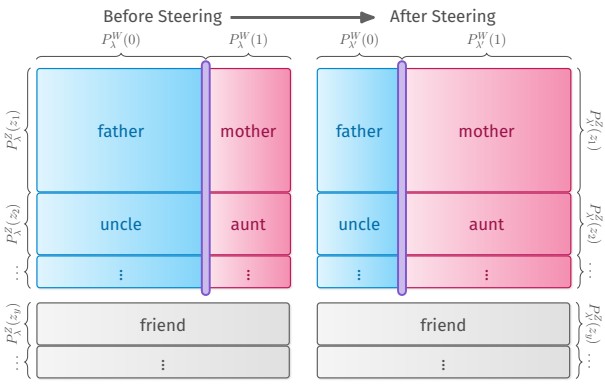

Figure 3. Illustration of an ideal steering process. The blue, red, and gray boxes represent partitions of the total probability mass. The goal is to shift only the boundary (purple bar) governing the concept distribution $P_\lambda^W$, while preserving the vertical partitions defined by the off-target distribution $P_\lambda^Z$.

the next token with the following probabilities in the left plot of Figure 3. If we intervene on the concept `male` $\Rightarrow$ `female` while preserving all off-target semantic concepts, the ideal resulting probabilities should be the right plot of Figure 3. In this ideal intervention, the probability mass is redistributed exclusively within relevant counterfactual pairs; e.g., the mass on "father" should move to "mother". Crucially, tokens that do not encode the targeted binary concept, such as "friend", should remain entirely unaffected.

To formalize this intuition, we partition the output space $\mathcal{Y}$ into a set of $n_W$ counterfactual pairs $\mathcal{Y}_W = \cup_{i=1}^{n_W} \{y_i^0, y_i^1\}$ corresponding to a binary concept $W \in \{0, 1\}$, and a set of neutral outputs $(\mathcal{Y}_W)^c$ that do not encode the concept. We define the *off-target space* $\mathcal{Z}_W = \{z_1, \ldots, z_{n_W}\} \cup \{z_y : y \in (\mathcal{Y}_W)^c\}$, where each $z_i$ represents the shared semantic attributes of the pair $(y_i^0, y_i^1)$. This allows us to capture the idea of distribution that mixes over concept-related and concept-irrelevant components:

**Definition 3.1** (Concept-Factorizable Distribution). A probability distribution $P_\lambda$ over $\mathcal{Y}$ is *concept-factorizable* with respect to $W$ if there exists a concept distribution $P_\lambda^W$ over $\{0, 1\}$ and an off-target distribution $P_\lambda^Z$ over $\mathcal{Z}_W$ such that

$$P_\lambda(y) = \begin{cases} P_\lambda^W(w) P_\lambda^Z(z_i) & \text{if } y = y_i^w \in \mathcal{Y}_W, \\ P_\lambda^Z(z_y) & \text{if } y \in (\mathcal{Y}_W)^c, \end{cases}$$

where $P_\lambda^Z$ is a valid probability distribution satisfying $\sum_{i=1}^{n_W} P_\lambda^Z(z_i) + \sum_{y \in (\mathcal{Y}_W)^c} P_\lambda^Z(z_y) = 1$. Under this factorization, $P(W = w \mid \lambda) = P_\lambda^W(w)$ for $w = 0, 1$.

In our running example, the on-target concept $W$ corresponds to the binary concept `male` $\Rightarrow$ `female`. The variable $z_1$ represents the shared semantic attribute "parent" (spanning the counterfactual pair "father" and "mother"),

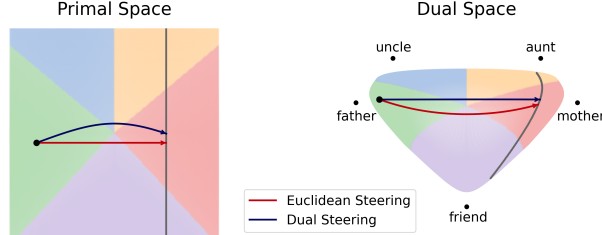

Figure 4. Illustration of Euclidean versus dual steering in the primal and dual spaces. Euclidean steering (red) drifts into a region where the unrelated token "friend" gains more probability mass, whereas dual steering (blue) remains within the region where the off-target distribution is preserved. The background color indicates the top-1 token at each point, e.g., green for "father" and purple for "friend".

while $z_{\text{friend}}$ represents the neutral token "friend." In Figure 3, the horizontal split (purple bar) defines the concept distribution, while the block heights represent the off-target distribution.

With this definition in hand, we can now prove that dual steering modifies the target concept while minimizing impact on the off-target distribution:

**Theorem 3.1** (Dual Steering with a Linear Probe). *Suppose there exists a linear probe $\beta_W$ for a binary concept $W$ satisfying (2). Given a context embedding $\lambda_0$ and a hyperplane $\Lambda_W(c) := \{\lambda : \beta_W^\top \lambda = c\}$, if a minimizer $\hat{\lambda} \in \arg\min_{\lambda \in \Lambda_W(c)} D_{\text{KL}}(P_{\lambda_0} \parallel P_\lambda)$ exists, we have*

$$\phi(\hat{\lambda}) = \phi(\lambda_0) + t\beta_W \quad \text{for some } t \in \mathbb{R}. \quad (4)$$

*Furthermore, if $P_\lambda$ is concept-factorizable with respect to $W$ for all $\lambda \in \Lambda_W(c) \cup \{\lambda_0\}$, then $\hat{\lambda}$ satisfies*

$$\hat{\lambda} \in \arg\min_{\lambda \in \Lambda_W(c)} D_{\text{KL}}(P_{\lambda_0}^Z \parallel P_\lambda^Z). \quad (5)$$

*Thus, dual steering identifies a minimizer $\hat{\lambda}$ on the hyperplane that best preserves the off-target distribution $P^Z$ while modifying the concept distribution $P^W$.*

Intuitively, as shown in Figure 4, dual steering remains constrained to the region preserving the off-target distribution. This behavior corresponds to the ideal intervention depicted in Figure 3: dual steering shifts the purple bar horizontally to alter the on-target concept, while maintaining the height of each block to preserve the off-target distributions.

### 3.2. Asymmetry of Steering

One might wonder if a symmetric property holds for Euclidean steering. Suppose instead we had a linear probe $\delta_W$ onto the dual space (i.e., $P(W = 1 \mid \lambda) = \sigma(\delta_W^\top \phi(\lambda) + \tilde{b}_W)$). In this case, $\delta_W$ is an element of the primal space. Such a representation might be constructed by, e.g., encoding pairs of inputs that vary by the target concept

($\lambda$("he is the") vs $\lambda$("she is the")) and taking $\delta_W$ to be the mean difference vector. Then it would be natural to identify the goal of steering as finding the reverse KL projection[2] onto the hyperplane $\Phi_W(c) := \{\phi \in \Phi : \delta_W^\top \phi = c\}$ in the dual space. In this case, similarly to (34) in the proof, we can use the concept-factorization assumption to decompose the reverse KL divergence as:

$$\sum_{i=1}^{n_W} P_\lambda^Z(z_i) \cdot D_{\text{KL}}\left(P_\lambda^W \parallel P_{\lambda_0}^W\right) + D_{\text{KL}}\left(P_\lambda^Z \parallel P_{\lambda_0}^Z\right) \quad (6)$$

The first term captures the on-target divergence and the second term captures the off-target divergence. Crucially, even if $D_{\text{KL}}\left(P_\lambda^W \parallel P_{\lambda_0}^W\right)$ is constant on the hyperplane, the total probability mass of the counterfactual pairs $\sum_{i=1}^{n_W} P_\lambda^Z(z_i)$ depends on $\lambda$ and cannot be ignored. Consequently, the projection does not generally preserve the off-target distribution $P^Z$. Instead, it tends to minimize the total probability mass of tokens in $\mathcal{Y}_W$ relative to the others. This leads to the "leakage" observed in practice, where Euclidean steering inadvertently shifts mass to unrelated tokens, thereby failing to maintain semantic invariance, as illustrated in Figure 4.

Note that, in practice, the concept-factorizability assumption in Definition 3.1 may not always hold. This assumption is used to prove Theorem 3.1, however, the difference between Euclidean and dual steering does not *require* factorization. This difference in steering behavior is due to the way that probability transfers between counterfactual pairs and other unrelated tokens with the different steering methods.

## 4. Practical Implementation of Dual Steering

We have established theoretically that dual steering effectively modifies the target concept while preserving off-target distributions. We now turn to how to implement dual steering in practice.

### 4.1. Feasibility Constraints and Rank-Deficiency

A key challenge here is that while the primal space $\Lambda$ is (largely) unconstrained, the dual space $\Phi$ is only a bounded convex set. For example, in the case of a fixed, finite set of items, each dual vector must be in the convex hull of those items.[3] This means that when we update $\phi' = \phi(\lambda_0) + t\beta_W$, we need to ensure that $\phi'$ remains within the interior of the convex hull of the unembedding vectors. Otherwise, there is no representation vector $\lambda_t$ so that $\phi(\lambda_t) = \phi'$.

To circumvent this issue, we trace the (non-linear) path in the primal space that corresponds to linear steering in the

---

[2]Recall from Proposition 2.1 that an $e$-geodesic corresponds to a minimizer of the reverse KL divergence.

[3]This is easy to see if we recall that the dual coordinate $\phi(\lambda) = \mathbb{E}[\gamma \mid \lambda]$ is a convex combination of the unembedding vectors $\gamma_y$ weighted by the softmax probabilities $P_\lambda$.

dual space. Using a first-order Taylor expansion, we can approximate the change in dual coordinates via the Hessian of the log-normalizer:

$$\nabla A(\lambda') - \nabla A(\lambda) \approx \nabla^2 A(\lambda)(\lambda' - \lambda).$$

The incremental update in the primal space is then given by taking a step $\Delta\lambda$ such that

$$\nabla^2 A(\lambda)\Delta\lambda = \varepsilon\beta_W$$

for infinitesimal $\varepsilon > 0$. Now, in the softmax case, the Hessian $\nabla^2 A(\lambda)$ corresponds to the covariance matrix of the unembedding vectors under the softmax distribution $P_\lambda$:

$$\nabla^2 A(\lambda) = \text{Cov}[\gamma \mid \lambda].$$

All together, this gives us a way to compute a primal update $\lambda' = \lambda + \Delta\lambda$ that corresponds to a small step in the dual space along the concept direction $\beta_W$ by solving the linear system:

$$\text{Cov}[\gamma \mid \lambda]\Delta\lambda = \varepsilon\beta_W. \quad (7)$$

Then, we can choose a small step size and iteratively apply this update to trace out the dual steering path.

However, this approach breaks when the Hessian is (numerically) rank-deficient. If the concept direction $\beta_W$ lies outside the column space of the Hessian, the linear system in (7) has no solution. Geometrically, this signifies that the dual coordinate has encountered a "wall" at the boundary of the convex hull $\Phi$. This happens frequently in practice. The underlying reason is that when the softmax is highly concentrated on a few tokens, the covariance matrix (the Hessian) is low-rank, as it only captures variation among those few tokens. For example, if $\beta_W$ represents a binary concept male $\Rightarrow$ female but the starting representation $\lambda_0$ only assigns probability to tokens like "father" and "uncle" then the column space of the Hessian will not contain the direction from, e.g., "father" to "mother". In this case, the direct steering update for the male to female direction is ill-defined.

### 4.2. Robust Steering via Regularized Newton Updates

To overcome this singularity, we employ a regularized Newton method as detailed in Algorithm 1. We introduce a regularization parameter $\alpha > 0$ to the covariance matrix:

$$(\text{Cov}[\gamma \mid \lambda] + \alpha I_d)v = \varepsilon\beta_W.$$

This ensures the matrix is full-rank and invertible. The resulting movement in the dual space follows:

$$\phi(\lambda') - \phi(\lambda) \approx \varepsilon\text{Cov}[\gamma \mid \lambda](\text{Cov}[\gamma \mid \lambda] + \alpha I_d)^{-1}\beta_W.$$

The behavior of this update is twofold. When $\beta_W$ lies within the range of the Hessian, the regularization effect is

---

**Algorithm 1** Dual Steering via Regularized Newton

---

**Input:** initial primal point $\lambda_0 \in \Lambda$, concept direction $\beta_W$
**Parameters:** regularization $\alpha$, steps $T$, step size $\eta$
**for** $t = 0$ **to** $T - 1$ **do**
  Compute covariance: $\Sigma_t \leftarrow \mathrm{Cov}[\gamma \mid \lambda_t]$
  Regularize: $\Sigma_t^{\mathrm{reg}} \leftarrow \Sigma_t + \alpha I_d$
  Solve linear system: $\Sigma_t^{\mathrm{reg}} v = \beta_W$
  Normalize and step: $\lambda_{t+1} \leftarrow \lambda_t + \eta \dfrac{v}{\|v\|_2}$
**end for**
**Output:** path $\{\lambda_t\}_{t=0}^{T}$

---

negligible for small $\alpha$, allowing for dual steering along the concept direction. Conversely, when $\beta_W$ resides in the null space—where $P_\lambda$ is so concentrated that the dual coordinate hits the boundary of $\Phi$—the regularized step $v$ nudges the distribution toward regions of higher entropy. This increases the local variance of the concept $W$, eventually bringing $\beta_W$ back into the range of the Hessian.[4]

In practice, to solve the linear program at each step efficiently, we use the conjugate gradient method, as provided in Algorithm 2. The new computational complexity is $\mathcal{O}(nkd)$, where $k$ is the vocabulary size, $d$ is the dimensionality of the embedding space, and $n$ is the number of conjugate gradient iterations (e.g., $n = 20$). This is faster than the naive $\mathcal{O}(kd^2 + d^3)$ algorithm, which directly computes the covariance matrix and its inverse, while yielding the same results.

# 5. Experiments

We now turn to the empirical evaluation of the dual steering method. To analyze LLM behavior, we utilize Gemma-3-4B (Kamath et al., 2025) with embeddings of contexts sampled from AllenAI C4 (Raffel et al., 2020). We implement steering for several binary concepts, such as verb $\Rightarrow$ third-person, verb $\Rightarrow$ ing, verb $\Rightarrow$ past, and English $\Rightarrow$ French. For vision-language behavior, we employ MetaClip-2 (Chuang et al., 2025). Our evaluation utilizes images from synthetic object datasets (e.g., "blue circles + green squares") and the COCO dataset (Lin et al., 2014), where the image embeddings serve as $\gamma_y$ for the softmax distribution in (1). Specifically, we implement steering for concepts such as blue $\Rightarrow$ red for the synthetic dataset and dog $\Rightarrow$ cat for the COCO dataset.

It is worth noting that off-target concepts in image models behave slightly differently than in LLMs. For instance, when steering the concept dog $\Rightarrow$ cat, an image containing a dog and a bicycle, and one containing a cat and a bicycle, constitute a counterfactual pair. However, an im-

---

[4]See Section C.1 for details.

---

age containing both a dog and a cat does not have a valid counterfactual pairing since it contains both attributes; consequently, it should be treated as a neutral example whose probability should remain unaffected by dual steering.

We construct our test probes using sets of representation vectors $\{\lambda_i^0\}_{i=1}^n$ and $\{\lambda_i^1\}_{i=1}^n$ where each element expresses the base or target attribute (e.g., "He is the" in one set, "She is the" in the other). We define two directions, the primal mean difference (Primal MD)

$$\beta_W^\Lambda = \frac{1}{n} \sum_i \lambda_i^1 - \frac{1}{n} \sum_i \lambda_i^0,$$

and the dual mean difference (Dual MD)

$$\beta_W^\Phi = \frac{1}{n} \sum_i \phi(\lambda_i^1) - \frac{1}{n} \sum_i \phi(\lambda_i^0).$$

As shown in Figure 7, both directions effectively serve as probes. Then, using a test set of representation vectors that express the base attribute, we perform both Euclidean and dual steering along each direction. See Section B for further details.

## 5.1. Metrics

We need to measure both the success of steering the target concept and the preservation of off-target concepts. The target concept probability $P_\lambda^W(1)$ can be measured as

$$P(W = 1 \mid \lambda_t) = \frac{\sum_{i=1}^{n_W} P(y_i^1 \mid \lambda_t)}{\sum_{i=1}^{n_W} P(y_i^0 \mid \lambda_t) + \sum_{i=1}^{n_W} P(y_i^1 \mid \lambda_t)},$$

where $\lambda_t$ is the steered representation vector at step $t$. We expect all steering methods to increase this probability to $1$.

Measuring off-target drift is more subtle. We consider three metrics based on the off-target distribution $P_\lambda^Z$ defined in Definition 3.1. First, we measure the total probability mass assigned to the counterfactual pairs during steering:

$$\sum_{i=1}^{n_W} P_{\lambda_t}^Z(z_i) = \sum_{i=1}^{n_W} P(y_i^0 \mid \lambda_t) + \sum_{i=1}^{n_W} P(y_i^1 \mid \lambda_t).$$

We evaluate whether Euclidean steering assigns lower probability mass to these counterfactual pairs than dual steering, as conjectured in Section 3.2. Second, we calculate the KL divergence of the off-target distributions between the initial and steered parameters:

$$D_{\mathrm{KL}}\left(P_{\lambda_0}^Z \parallel P_{\lambda_t}^Z\right) = \sum_{z \in \mathcal{Z}_W} P_{\lambda_0}^Z(z) \log \frac{P_{\lambda_0}^Z(z)}{P_{\lambda_t}^Z(z)}.$$

A small value indicates that the off-target distribution is well-preserved. However, even with a low KL divergence, the relative ranking of off-target components $z \in \mathcal{Z}_W$ may

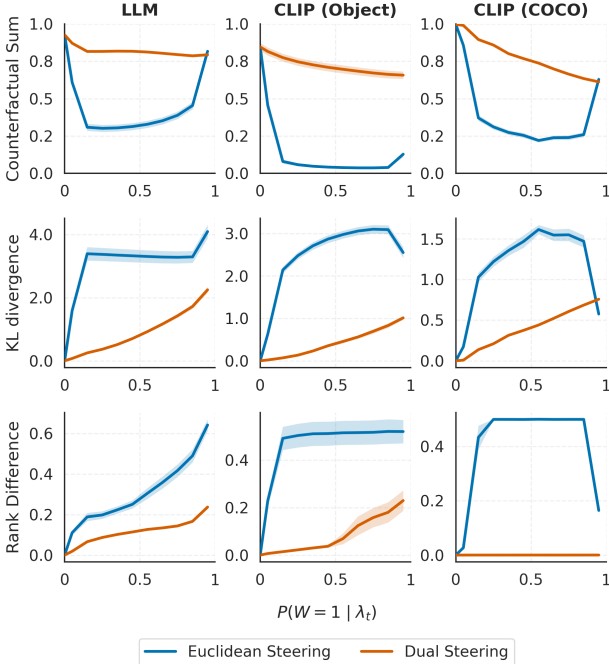

*Figure 5.* **Dual steering (red) consistently preserves off-target distributions better than Euclidean steering (blue), while both boost the target concept probability.** We plot three robustness metrics (y-axes) against the target concept probability (x-axis) achieved via steering along the dual mean difference. As the target concept probability approaches 1 (moving right), Euclidean steering degrades the off-target distribution, whereas dual steering maintains it. **Columns:** Tasks include LLM steering for `English ⇒ French` (left), CLIP on synthetic objects for `yellow ⇒ green` (middle), and CLIP on real images (COCO) for `carrot ⇒ broccoli` (right). **Rows:** The top row shows the total probability mass on counterfactual pairs (constant is better). The middle and bottom rows show the KL divergence and rank difference of off-target distributions (lower is better). Lines represent the mean, and shading indicates the standard error of the mean (SEM) across test contexts.

shift significantly. Therefore, we also measure the weighted sum of the inverse rank differences:

$$
\sum_{z \in \mathcal{Z}_W} P_{\lambda_0}^Z(z) \cdot \left| \frac{1}{\text{ranking}_{P_{\lambda_t}^Z}(z)} - \frac{1}{\text{ranking}_{P_{\lambda_0}^Z}(z)} \right|.
$$

A low value indicates that the ranking of off-target components remains stable during steering.

## 5.2. Results

Example steering results are shown in Figure 1. As expected, Figure 5 demonstrates that both Euclidean and dual steering successfully promote the target concept. However, Euclidean steering tends to distort the off-target distribution, often assigning significant probability mass to unrelated tokens during intermediate steps. As a result, dual steering outperforms Euclidean steering across all three robustness

metrics.

Additional steering results involving a broader range of concepts and directions are provided in Section C.3. These results show that dual steering consistently maintains superior performance over Euclidean steering in preserving off-target concepts. This holds regardless of whether Primal MD or Dual MD is employed as the linear probe.

### 5.3. Discussion

**When does Euclidean steering work well?** Euclidean steering occasionally succeeds in preserving off-target distributions. This typically occurs when the "counterfactual sum" $\sum_{i=1}^{n_W} P_{\lambda_t}^Z(z_i)$ remains relatively constant while a linear probe is added in the primal space. This stability makes the entire first term in (6) constant; in this regime, the second term—the KL divergence of the off-target distribution—is effectively minimized by Euclidean steering.

For example, consider a "base" distribution that concentrates almost all of its probability mass on tokens in counterfactual pairs. If we employ a suitable concept direction,[5] Euclidean steering will shift probability mass among the counterfactual tokens rather than leaking it toward neutral ones.

This condition is often met when using the Primal MD as the concept direction. Figures 9 and 10 show that Euclidean steering with the Primal MD preserves counterfactual sums and thus inherently minimizes off-target divergence across multiple concepts. However, even when KL divergence is minimized, the relative ranking of off-target components can shift significantly, as illustrated in Figure 11. Consequently, dual steering remains the more robust choice for preserving the off-target distribution.

**Probing Assumption** Steering with a linear probe is significantly impacted by the quality of the probe. In Theorem 3.1, we assume the concept probability $P(W = 1 \mid \lambda)$ is constant across the entire hyperplane $\Lambda_W(c)$. This formalizes the idea that the probe sufficiently represents the target concept without any off-target entanglement. This is a stringent requirement. In practice, probes are trained on finite datasets, and we have no guarantee that this probability invariance assumption holds when moving away from the training distribution. We provide an experimental illustration and further explanation of this phenomenon in Section C.2. However, even when this assumption is violated, empirical results show that dual steering still outperforms Euclidean steering in preserving off-target distributions.

---

[5]e.g., a concept direction $\beta_W$ satisfying $\beta_W^\top \gamma(y) > \beta_W^\top \gamma(y')$ for any $y = y_i^w \in \mathcal{Y}_W$ and $y' \in (\mathcal{Y}_W)^c$.

## 6. Discussion and Related Works

**Representation Geometry**   A long line of work has studied the geometric structure of representations learned by neural networks, especially in the context of word embeddings (Bengio et al., 2013; Mikolov et al., 2013a; Pennington et al., 2014; Arora et al., 2016; 2018). This work has been extended to modern LLMs exploring linear representations (Elhage et al., 2021; Marks & Tegmark, 2023; Hendel et al., 2023; Tigges et al., 2023; Nanda et al., 2023; Park et al., 2024; Arditi et al., 2024; Jain et al., 2024; Jiang et al., 2024), as well as polytopes, manifolds, and other geometric structures (Black et al., 2022; Elhage et al., 2022; Gurnee & Tegmark, 2024; Engels et al., 2024; Robinson et al., 2024; Park et al., 2025; Wollschläger et al., 2025; Modell et al., 2025; Robinson et al., 2025; Dooms & Gauderis, 2025; Gurnee et al., 2026). Related analysis has been done on multi-modal models; of note is work on interpreting the representations learned by CLIP models (Radford et al., 2021; Merullo et al., 2022; Gandelsman et al., 2023; 2024). The present work is the first to explore the interplay between information geometry and the linear representation hypothesis.

**Riemannian & Information Geometry**   Most closely related to our work is the exploration of Riemannian geometry in generative models (Arvanitidis et al., 2017; Shao et al., 2018; Arvanitidis et al., 2020; Yu et al., 2025), especially through the lens of information geometry (Amari, 2016; Nielsen, 2020; Arvanitidis et al., 2022). In particular, Arvanitidis et al. (2022) use Riemannian Geometry to understand the geometry in the latent space of VAE. Focusing on the decoder distribution, they pull back the Fisher-Rao metric in the parameter space to the latent space and find the shortest path (geodesic with respect to Levi-Civita connection). By contrast, we focus on the parameter space of softmax with the dual ($e$- and $m$-) connections, which aligns with the linear representation hypothesis for probing and steering. Volpi & Malago (2021) also study the information geometry of word embeddings learned with a contrastive objective, focusing on similarity between word embeddings, whereas our work focuses on steering and probing softmax models.

**Steering**   A line of model steering work in LLMs has focused on using the difference in representations in a model's hidden layers to predictably alter model behavior (Li et al., 2023; Liu et al., 2023; Turner et al., 2023; Zou et al., 2023; Rimsky et al., 2024). Follow-up work has examined the robustness of steering interventions, noting that model steering directions are often ineffective, can cause off-target concept drift, and are sensitive to the choice of steering magnitude (Tan et al., 2024; Pres et al., 2024; Wu et al., 2025). Concurrent work (Sarfati et al., 2026; Wurgaft et al., 2026) focuses on learning manifolds in language models' representation spaces and steering along these manifolds to avoid off-target drift. Another line of work focuses on editing model representations to erase concepts (Bolukbasi et al., 2016; Vargas & Cotterell, 2020; Belrose et al., 2023; Ravfogel et al., 2022). Most closely related is Singh et al. (2024), which discusses steering a concept while minimally perturbing $L^2$ distance; we focus on the KL divergence in the output distribution as our notion of distance.

Other steering work focuses on examining logit differences (Park et al., 2024; 2025). They argue that it is possible to intervene on a specific concept by adding a direction induced by a "causal inner product." For instance, adding the estimated direction for the binary concept `male` $\Rightarrow$ `female` might change the logit difference between 'king' and 'queen' while leaving the logit difference between 'king' and 'King' unaltered. However, logit differences are not probabilities. Even if specific logit differences remain constant, the resulting probabilities can vary significantly due to the contributions of other logits through the softmax operation. Furthermore, the scale of logit differences is not directly proportional to the magnitude of probability changes. Therefore, to evaluate the effect of steering more rigorously, we focus on changes in the model's probability distribution.

**Future Directions**   Empirically, we have focused on the parameter space of the softmax layer, which directly governs the output distribution. This is experimentally convenient for measuring the geometry, constructing probes, and evaluating the steering. However, in practice, steering is often applied to intermediate layers within the model. Understanding how the geometry of the softmax layer, and of attention layers, influences the geometry of these intermediate layers is an important direction for future research.

## Acknowledgements

This work is supported by ONR grant N00014-23-1-2591 and Open Philanthropy.

## Impact Statement

This paper presents work whose goal is to advance the field of Machine Learning. There are many potential societal consequences of our work, none of which we feel must be specifically highlighted here.

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

# A. Proofs

## A.1. Proof of Proposition 2.1

**Proposition 2.1** (Interpolation as KL Minimization). *The primal interpolation $\lambda_t$ minimizes a weighted sum of **reverse** KL divergences:*

$$\lambda_t \in \operatorname*{argmin}_{\lambda \in \Lambda}(1-t)D_{\mathrm{KL}}\left(P_\lambda \parallel P_{\lambda_0}\right) + tD_{\mathrm{KL}}\left(P_\lambda \parallel P_{\lambda_1}\right),$$

*whereas the dual interpolation $\phi_t$ minimizes a weighted sum of **forward** KL divergences:*

$$\phi_t \in \operatorname*{argmin}_{\phi \in \Phi}(1-t)D_{\mathrm{KL}}\left(P_{\lambda_0} \parallel P_\phi\right) + tD_{\mathrm{KL}}\left(P_{\lambda_1} \parallel P_\phi\right).$$

*Proof.* Following Proposition 1 in Banerjee et al. (2005), we provide a direct proof for completeness. While the original result assumes strict convexity of the log-normalizer $A$ to ensure a unique solution, we show that the arithmetic mean is a minimizer even when $A$ is not strictly convex.

(Primal Interpolation) We express the weighted sum of reverse KL divergences as

$$f(\lambda) := (1-t)D_{\mathrm{KL}}\left(P_\lambda \parallel P_{\lambda_0}\right) + tD_{\mathrm{KL}}\left(P_\lambda \parallel P_{\lambda_1}\right) \tag{8}$$

$$= (1-t)(A(\lambda_0) - A(\lambda) - \nabla A(\lambda)^\top(\lambda_0 - \lambda)) + t(A(\lambda_1) - A(\lambda) - \nabla A(\lambda)^\top(\lambda_1 - \lambda)) \tag{9}$$

$$= (1-t)A(\lambda_0) + tA(\lambda_1) - A(\lambda) - \nabla A(\lambda)^\top\left((1-t)\lambda_0 + t\lambda_1 - \lambda\right) \tag{10}$$

$$= \text{const} - A(\lambda) - \nabla A(\lambda)^\top(\lambda_t - \lambda), \tag{11}$$

where $\lambda_t = (1-t)\lambda_0 + t\lambda_1$ denotes the primal interpolation. For any $\lambda' \in \Lambda$,

$$f(\lambda') - f(\lambda_t) = -A(\lambda') - \nabla A(\lambda')^\top(\lambda_t - \lambda') + A(\lambda_t) + \nabla A(\lambda_t)^\top(\lambda_t - \lambda_t) \tag{12}$$

$$= A(\lambda_t) - A(\lambda') - \nabla A(\lambda')^\top(\lambda_t - \lambda') \tag{13}$$

$$= D_{\mathrm{KL}}\left(P_{\lambda'} \parallel P_{\lambda_t}\right) \geq 0. \tag{14}$$

Therefore, the primal interpolation $\lambda_t$ minimizes the weighted sum of reverse KL divergences.

(Dual Interpolation) For any $\phi \in \Phi$, there exists $\lambda \in \Lambda$ such that $\phi = \nabla A(\lambda)$. We express the weighted sum of forward KL divergences as

$$f(\lambda) := (1-t)D_{\mathrm{KL}}\left(P_{\lambda_0} \parallel P_\phi\right) + tD_{\mathrm{KL}}\left(P_{\lambda_1} \parallel P_\phi\right) \tag{15}$$

$$= (1-t)D_{\mathrm{KL}}\left(P_{\lambda_0} \parallel P_\lambda\right) + tD_{\mathrm{KL}}\left(P_{\lambda_1} \parallel P_\lambda\right) \tag{16}$$

$$= (1-t)(A(\lambda) - A(\lambda_0) - \nabla A(\lambda_0)^\top(\lambda - \lambda_0)) + t(A(\lambda) - A(\lambda_1) - \nabla A(\lambda_1)^\top(\lambda - \lambda_1)) \tag{17}$$

$$= A(\lambda) - ((1-t)A(\lambda_0) + tA(\lambda_1)) - \phi_t^\top\lambda + (1-t)\nabla A(\lambda_0)^\top\lambda_0 + t\nabla A(\lambda_1)^\top\lambda_1 \tag{18}$$

$$= A(\lambda) - \phi_t^\top\lambda + \text{const}, \tag{19}$$

where $\phi_t = (1-t)\nabla A(\lambda_0) + t\nabla A(\lambda_1)$ denotes the dual interpolation. Differentiating with respect to $\lambda$,

$$\nabla_\lambda f(\lambda) = \nabla A(\lambda) - \phi_t = \phi - \phi_t. \tag{20}$$

The second-order derivative is given by the Hessian of the log-normalizer:

$$\nabla_\lambda^2 f(\lambda) = \nabla^2 A(\lambda) \succeq 0, \tag{21}$$

which is positive semi-definite. Therefore, the weighted dual interpolation $\phi_t$ minimizes the weighted sum of forward KL divergences. $\square$

## A.2. Proof of Theorem 3.1

**Theorem 3.1** (Dual Steering with a Linear Probe). *Suppose there exists a linear probe $\beta_W$ for a binary concept $W$ satisfying (2). Given a context embedding $\lambda_0$ and a hyperplane $\Lambda_W(c) := \{\lambda : \beta_W^\top \lambda = c\}$, if a minimizer $\hat{\lambda} \in \arg\min_{\lambda \in \Lambda_W(c)} D_{\mathrm{KL}}(P_{\lambda_0} \| P_\lambda)$ exists, we have*

$$\phi(\hat{\lambda}) = \phi(\lambda_0) + t\beta_W \quad \text{for some } t \in \mathbb{R}. \tag{4}$$

*Furthermore, if $P_\lambda$ is concept-factorizable with respect to $W$ for all $\lambda \in \Lambda_W(c) \cup \{\lambda_0\}$, then $\hat{\lambda}$ satisfies*

$$\hat{\lambda} \in \underset{\lambda \in \Lambda_W(c)}{\arg\min}\, D_{\mathrm{KL}}\left(P_{\lambda_0}^Z \| P_\lambda^Z\right). \tag{5}$$

*Thus, dual steering identifies a minimizer $\hat{\lambda}$ on the hyperplane that best preserves the off-target distribution $P^Z$ while modifying the concept distribution $P^W$.*

*Proof.* (Proof for (4)) This result is grounded on the Projection Theorem in information geometry (Amari, 2016). However, for completeness, we provide a direct proof here. We represent the hyperplane $\Lambda_W(c)$ with a basis $\{v_1, \ldots, v_{d-1}\}$ for the null space of $\beta_W^\top$. Any $\lambda \in \Lambda_W(c)$ can be written as:

$$\lambda(\alpha) = c_0 + \sum_{i=1}^{d-1} \alpha_i v_i, \quad \alpha = (\alpha_1, \ldots, \alpha_{d-1}) \in \mathbb{R}^{d-1}, \tag{22}$$

where $\beta_W^\top c_0 = c$. We express the KL divergence between $\lambda_0$ and $\lambda(\alpha) \in \Lambda_W(c)$ as a function of $\alpha$:

$$f(\alpha) = D_{\mathrm{KL}}\left(P_{\lambda_0} \| P_{\lambda(\alpha)}\right) \tag{23}$$

$$= A(\lambda(\alpha)) - A(\lambda_0) - \nabla A(\lambda_0)^\top (\lambda(\alpha) - \lambda_0) \tag{24}$$

$$= A\left(c_0 + \sum_{i=1}^{d-1} \alpha_i v_i\right) - \phi(\lambda_0)^\top \left(c_0 + \sum_{i=1}^{d-1} \alpha_i v_i\right) + \text{const.} \tag{25}$$

The first-order optimality condition for a minimizer $\hat{\lambda} = \lambda(\hat{\alpha}) \in \Lambda_W(c)$ is given by:

$$\left.\frac{\partial}{\partial \alpha_i} f(\alpha)\right|_{\alpha = \hat{\alpha}} = \left(\nabla A(\lambda(\hat{\alpha})) - \phi(\lambda_0)\right)^\top v_i = 0, \quad i = 1, \ldots, d-1. \tag{26}$$

Since $\nabla A(\lambda(\hat{\alpha})) = \phi(\hat{\lambda})$, $\phi(\hat{\lambda}) - \phi(\lambda_0)$ is orthogonal to all basis vectors $\{v_1, \ldots, v_{d-1}\}$ of the hyperplane. Thus, $\phi(\hat{\lambda}) - \phi(\lambda_0)$ is parallel to $\beta_W$.

(Proof for (5)) For a given context embedding $\lambda_0$ and any $\lambda \in \Lambda_W(c)$, the KL divergence is decomposed as follows:

$$D_{\mathrm{KL}}(P_{\lambda_0} \| P_\lambda) = \sum_{y \in \mathcal{Y}} P_{\lambda_0}(y) \log \frac{P_{\lambda_0}(y)}{P_\lambda(y)} \tag{27}$$

$$= \sum_{i=1}^{n_W} \sum_w P_{\lambda_0}(y_i^w) \log \frac{P_{\lambda_0}(y_i^w)}{P_\lambda(y_i^w)} + \sum_{y \in (\mathcal{Y}_W)^c} P_{\lambda_0}(y) \log \frac{P_{\lambda_0}(y)}{P_\lambda(y)}. \tag{28}$$

Since $P_{\lambda_0}$ and $P_\lambda$ are concept-factorizable with respect to $W$, the second term becomes

$$\sum_{y \in (\mathcal{Y}_W)^c} P_{\lambda_0}(y) \log \frac{P_{\lambda_0}(y)}{P_\lambda(y)} = \sum_{y \in (\mathcal{Y}_W)^c} P_{\lambda_0}^Z(z_y) \log \frac{P_{\lambda_0}^Z(z_y)}{P_\lambda^Z(z_y)}, \tag{29}$$

and the first term becomes

$$\sum_{i=1}^{n_W} \sum_w P_{\lambda_0}(y_i^w) \log \frac{P_{\lambda_0}(y_i^w)}{P_\lambda(y_i^w)} = \sum_{i=1}^{n_W} \sum_w P_{\lambda_0}^W(w) P_{\lambda_0}^Z(z_i) \log \frac{P_{\lambda_0}^W(w) P_{\lambda_0}^Z(z_i)}{P_\lambda^W(w) P_\lambda^Z(z_i)} \tag{30}$$

$$= \sum_{i=1}^{n_W} \sum_w P_{\lambda_0}^W(w) P_{\lambda_0}^Z(z_i) \log \frac{P_{\lambda_0}^W(w)}{P_\lambda^W(w)} + \sum_{i=1}^{n_W} \sum_w P_{\lambda_0}^W(w) P_{\lambda_0}^Z(z_i) \log \frac{P_{\lambda_0}^Z(z_i)}{P_\lambda^Z(z_i)} \tag{31}$$

$$= \sum_{i=1}^{n_W} P_{\lambda_0}^Z(z_i) \cdot \sum_w P_{\lambda_0}^W(w) \log \frac{P_{\lambda_0}^W(w)}{P_\lambda^W(w)} + \sum_{i=1}^{n_W} P_{\lambda_0}^Z(z_i) \log \frac{P_{\lambda_0}^Z(z_i)}{P_\lambda^Z(z_i)}, \tag{32}$$

because $\sum_w P_{\lambda_0}^W(w) = 1$. Together, we have

$$D_{\mathrm{KL}}\left(P_{\lambda_0} \parallel P_\lambda\right) = \sum_{i=1}^{n_W} P_{\lambda_0}^Z(z_i) \cdot D_{\mathrm{KL}}\left(P_{\lambda_0}^W \parallel P_\lambda^W\right) + \sum_{z \in \mathcal{Z}_W} P_{\lambda_0}^Z(z) \log \frac{P_{\lambda_0}^Z(z)}{P_\lambda^Z(z)} \tag{33}$$

$$= \sum_{i=1}^{n_W} P_{\lambda_0}^Z(z_i) \cdot D_{\mathrm{KL}}\left(P_{\lambda_0}^W \parallel P_\lambda^W\right) + D_{\mathrm{KL}}\left(P_{\lambda_0}^Z \parallel P_\lambda^Z\right). \tag{34}$$

On the hyperplane $\Lambda_W(c)$, the value of the linear probe is fixed to $c + b_W$, meaning that the concept distribution $P_\lambda^W$ is constant for all $\lambda \in \Lambda_W(c)$. Consequently, the first term $\sum_{i=1}^{n_W} P_{\lambda_0}^Z(z_i) \cdot D_{\mathrm{KL}}\left(P_{\lambda_0}^W \parallel P_\lambda^W\right)$ is independent of $\lambda$ within this hyperplane. The optimization problem thus simplifies as follows:

$$\hat{\lambda} \in \underset{\lambda \in \Lambda_W(c)}{\arg\min} D_{\mathrm{KL}}\left(P_{\lambda_0} \parallel P_\lambda\right) \tag{35}$$

$$= \underset{\lambda \in \Lambda_W(c)}{\arg\min} \left(\mathrm{const} + D_{\mathrm{KL}}\left(P_{\lambda_0}^Z \parallel P_\lambda^Z\right)\right) \tag{36}$$

$$= \underset{\lambda \in \Lambda_W(c)}{\arg\min} D_{\mathrm{KL}}\left(P_{\lambda_0}^Z \parallel P_\lambda^Z\right). \tag{37}$$

$$\square$$

## B. Experimental Details

### B.1. Dataset

#### B.1.1. LARGE LANGUAGE MODELS (LLMs)

To analyze LLMs, we first construct counterfactual pairs of tokens. Using contexts that typically precede verb tokens (e.g., "I don't want to"), we sweep all tokens generated via Top-$p$ sampling. We then utilize the Claude API to identify the base form of each verb and generate token pairs that differ across specific binary concepts, such as `verb` $\Rightarrow$ `third-person`, `verb` $\Rightarrow$ `ing`, `verb` $\Rightarrow$ `past`, and `English` $\Rightarrow$ `French`. This mapping consists of over 300 token pairs; while some rare tokens might be omitted from this dictionary, their absence does not significantly undermine the results.

Next, we sample 10,000 text sequences from the C4 dataset and collect the context embeddings for the first 256 tokens of each sequence. These embeddings are extracted from the final transformer layer, while unembedding vectors are taken directly from the weight matrix of the softmax layer. For each binary concept, we identify two groups of context embeddings where the Top-3 predicted tokens belong to either the base or target group of the counterfactual pairs, provided the cumulative probability of these tokens is at least 0.7. Finally, this collection is partitioned into training and test sets. Notably, the training data is derived from the natural distribution of the C4 dataset rather than from specifically constructed counterfactual contexts.

#### B.1.2. CLIP MODELS

For the CLIP models, we construct the entire image vocabulary using two datasets: the COCO dataset and a synthetic object dataset featuring compositions of colors and shapes. We generate the synthetic dataset using GPT-Image-1. Random samples are manually verified, with no errors found. Since the features in these datasets are distinct and isolated, counterfactual pairs

*Table 1.* Frobenius norm of the difference between the top-$K$ approximation and the full Fisher information matrix along the steering trajectory. The approximation error approaches zero at $K = 20{,}000$.

| TOP-$K$ | STEP 0 | STEP 5 | STEP 10 | STEP 50 | STEP 100 |
|---|---|---|---|---|---|
| 1 | 0.2031 | 0.2031 | 0.2031 | 0.2926 | 0.3321 |
| 20 | 0.0120 | 0.0133 | 0.0151 | 0.0006 | 0.0000 |
| 2000 | 0.0001 | 0.0002 | 0.0002 | 0.0000 | 0.0000 |
| 20000 | 0.0000 | 0.0000 | 0.0000 | 0.0000 | 0.0000 |
| 200000 | 0.0000 | 0.0000 | 0.0000 | 0.0000 | 0.0000 |

are well-defined. For example, for the concept `circles ⇒ triangles`, pairs include ("blue circles", "blue triangles") and ("red circles and yellow squares", "red triangles and yellow squares").

For each binary concept, we generate two groups of captions (contexts) incorporating various co-occurring features and a set of prefixes, such as "a photo of". We use the CLIP text and image encoders to obtain the context embeddings and unembedding vectors, respectively. Because CLIP embeddings are normalized and scaled by a temperature parameter during training, we re-apply this temperature factor (from the final training step) to ensure the embeddings are correctly scaled for the softmax distribution. Following the LLM approach, we split the context embedding dataset into training and test sets.

### B.2. Steering

For both the LLM and CLIP model, we compute the linear probes (primal and dual mean differences) using the training dataset. We then perform Euclidean and dual steering along each linear probe, applied to the test set context embeddings. For dual steering, we employ the regularized Newton method described in Algorithm 1 with a tuned regularization parameter $\alpha$ (specifically, $\alpha = 5 \times 10^{-3}$ for both models). We iterate for a sufficient number of steps to ensure the target concept probability $P_{\lambda_t}^W(1)$ converges to approximately 1. Specifically, the process is terminated once the target concept probability $P_{\lambda_t}^W(1)$ first reaches 0.9999. Beyond this threshold, both steering processes typically cause the distribution to collapse onto a single token with a probability of one. As our objective is to analyze the interplay between the concept and off-target distributions throughout the steering trajectory, we exclude these edge cases.

Computationally, dual steering in Algorithm 1 is demanding as it requires updating the covariance matrix $\mathrm{Cov}[\gamma \mid \lambda_t]$ and its inverse at each step. The computational complexity of this step is $\mathcal{O}(kd^2 + d^3)$, where $d$ is the dimension of the representation and $k$ is the full vocabulary size. The large vocabulary size $k$ and representation dimension $d$ in the models (e.g., $k = 262$K and $d = 2$K for LLMs) make this step computationally prohibitive.

To address the large vocabulary size $k$, we approximate the covariance matrix using only the top-$K$ (e.g., $K = 20{,}000$) tokens at each step. This approximation remains highly accurate because the probability distribution is typically sparse; since the vast majority of the probability mass is concentrated on a few thousand tokens, the contribution of the long tail to the covariance structure is negligible. For example, as shown in Table 1, using top-$K$ tokens with $K \geq 20{,}000$ results in a very small approximation error.

With this top-$K$ approximation, we use the conjugate gradient (CG) method to solve the linear system $\Sigma_t^{\mathrm{reg}} v = \beta_W$ more efficiently at each step, as detailed in Algorithm 2. Rather than explicitly computing and inverting the full covariance matrix—which requires expensive matrix-matrix multiplications—the primary computational bottleneck becomes the matrix-vector product at each CG iteration. This reduces the total computational complexity to $\mathcal{O}(nKd)$, where $n$ is the number of CG iteration steps and $K \ll k$. In practice, we find that 20 iterations are sufficient to achieve a strong approximation of the solution.

### B.3. Metrics

When computing metrics along each path $\{\lambda_t\}_t$, the term $P(y_i \mid \lambda_t)$ represents the direct output probability from the softmax layer at step $t$. However, in the case of CLIP, a single token $y_i$ may correspond to multiple images. For instance, the token "blue circles" can serve as a caption for various distinct images. Consequently, $P(y_i \mid \lambda)$ is computed by aggregating the probabilities of all images in the vocabulary that are associated with the token $y_i$. For our main experiments, we use two images per token to maintain balance, while for Figures 1 and 2, we use a single image per token to ensure visual clarity.

For the off-target distribution $P_\lambda^Z$ in Definition 3.1, we define $P_\lambda^Z(z_i) = P_\lambda(y_i^0) + P_\lambda(y_i^1)$. In cases where multiple base

---

**Algorithm 2** Dual Steering via Conjugate Gradient

---

**Input:** initial primal $\lambda_0 \in \mathbb{R}^d$, concept direction $\beta_W \in \mathbb{R}^d$, unembedding matrix $G \in \mathbb{R}^{k \times d}$
**Parameters:** regularization $\alpha > 0$, outer steps $T$, step size $\eta$, top-$K$ tokens, CG iterations $M$, tolerance $\varepsilon$
**for** $t = 0$ **to** $T - 1$ **do**
    Compute logits and select top-$K$ indices: $\mathcal{K} \leftarrow \text{top-K}(G\lambda_t)$
    Compute distribution and mean embedding: $\tilde{p} \leftarrow \text{softmax}(G_{\mathcal{K}} \lambda_t)$, $\mu_t \leftarrow \tilde{p}^\top G_{\mathcal{K}}$
    Initialize: $v \leftarrow 0$, $r \leftarrow \beta_W$, $\rho \leftarrow r^\top r$, $p \leftarrow r$
    **for** $i = 0$ **to** $M - 1$ **do**
        Compute $\Sigma_t^{\text{reg}} p$: $q \leftarrow G_{\mathcal{K}}^\top \big( \tilde{p} \odot (G_{\mathcal{K}} \, p) \big) - \mu_t \, (\mu_t^\top p) + \alpha \, p$
        **if** $|p^\top q| < 10^{-10}$ **then**
            **break**
        **end if**
        $\alpha_i \leftarrow \dfrac{\rho}{p^\top q}$, $v \leftarrow v + \alpha_i \, p$, $r \leftarrow r - \alpha_i \, q$, $\rho' \leftarrow r^\top r$
        **if** $\sqrt{\rho'} < \varepsilon$ **then**
            **break**
        **end if**
        $p \leftarrow r + \dfrac{\rho'}{\rho} p$
        $\rho \leftarrow \rho'$
    **end for**
    Normalize and step: $\lambda_{t+1} \leftarrow \lambda_t + \eta \, \dfrac{v}{\|v\|_2}$
**end for**
**Output:** path $\{\lambda_t\}_{t=0}^T$

---

tokens $y_i^0$ correspond to a single target token $y_i^1$ (e.g., "purchase" and "buy" both mapping to the French "acheter" for the `English ⇒ French` concept), we compute $P_\lambda^Z(z_i)$ by aggregating the probabilities of all associated tokens into a single concept-level token $z_i$.

To ensure numerical stability when computing the KL divergence, we add an offset to the probabilities. This prevents the KL divergence from becoming unstable when probabilities approach zero. Since our analysis focuses on the behavior of the primary mass of the probability distribution, this offset does not meaningfully alter the results.

Calculating the difference in reverse ranks for every step is computationally prohibitive due to the large vocabulary size, particularly for LLMs. To address this, we select a subset of equally spaced steps along the path and identify the tokens constituting the Top 0.99 cumulative probability at each. We then take the union of these tokens to form a reduced vocabulary. The rank differences are computed by sorting only the tokens within this union, as omitted tokens have negligible probabilities that do not significantly impact the ranking.

Finally, to visualize the mean and standard error of the mean (SEM) across different test contexts, we utilize binning. We aggregate the metrics into discrete bins based on the target concept probability $P_\lambda^W(1)$. We first average the metrics within each bin for each individual path, and then compute the overall mean and SEM across all test contexts for each bin.

## B.4. Text Templates for CLIP Experiments

In this section, we present the text templates used in the CLIP experiments.

### B.4.1. OBJECT EXPERIMENTS

For the object experiments, we use the following text templates, yielding examples like "blue circles" and "a rendering of blue circles".

```
prefix_formats = [
    "",
    "a rendering of ",
    "a depiction of ",
    "an illustration of ",
```

```
        "a conceptual illustration of ",
        "A rendering of ",
        "A depiction of ",
        "An illustration of ",
        "A conceptual illustration of ",
        "Rendering of ",
        "Depiction of ",
        "Illustration of ",
        "Conceptual illustration of ",
    ]
```

### B.4.2. COCO EXPERIMENTS

For the experiments with COCO, we use the following text templates, yielding examples like "a photo of a dog" and "a picture of a dog".

```
prefix_formats = [
        "",
        "a photo of ",
        "a picture of ",
        "an image of ",
        "a photograph of ",
        "a snapshot of ",
        "A photo of ",
        "A picture of ",
        "An image of ",
        "A photograph of ",
        "A snapshot of ",
        "Photo of ",
        "Picture of ",
        "Image of ",
        "Photograph of ",
        "Snapshot of ",
    ]
```

## C. Additional Results

### C.1. Geometry of Dual Steering via Regularized Newton Updates

In Section 4, we discussed how regularization facilitates the implementation of dual steering. As previously noted, when the dual coordinate $\phi(\lambda)$ approaches the boundary of the feasible region $\Phi$, the regularized step $v$ shifts the distribution toward regions of higher entropy. This increases the local variance of the concept $W$, effectively bringing $\beta_W$ back within the range of the Hessian.

Figure 6 illustrates this behavior by plotting the cosine similarity between the actual dual step and the desired concept direction: $\cos(\phi(\lambda_{t+1}) - \phi(\lambda_t), \beta_W)$. In dual steering, the step is not perfectly aligned with the concept direction initially due to the influence of regularization. However, as steering progresses, the cosine similarity increases, indicating that dual steering effectively aligns with the concept direction in the dual space. In contrast, Euclidean steering maintains a lower cosine similarity throughout the process, as the dual step is a concept direction transformed by the Hessian.

Interestingly, for dual steering, the cosine similarity begins to decrease after reaching a peak. This occurs as the steering path again approaches the boundary of the convex hull $\Phi$. Geometrically, by leveraging the momentum provided by the concept direction, the steering path is able to "slide" along the low-dimensional faces of the convex hull boundary. Conversely, Euclidean steering typically moves through the deep interior of the convex hull, where irrelevant tokens are assigned non-negligible probabilities.

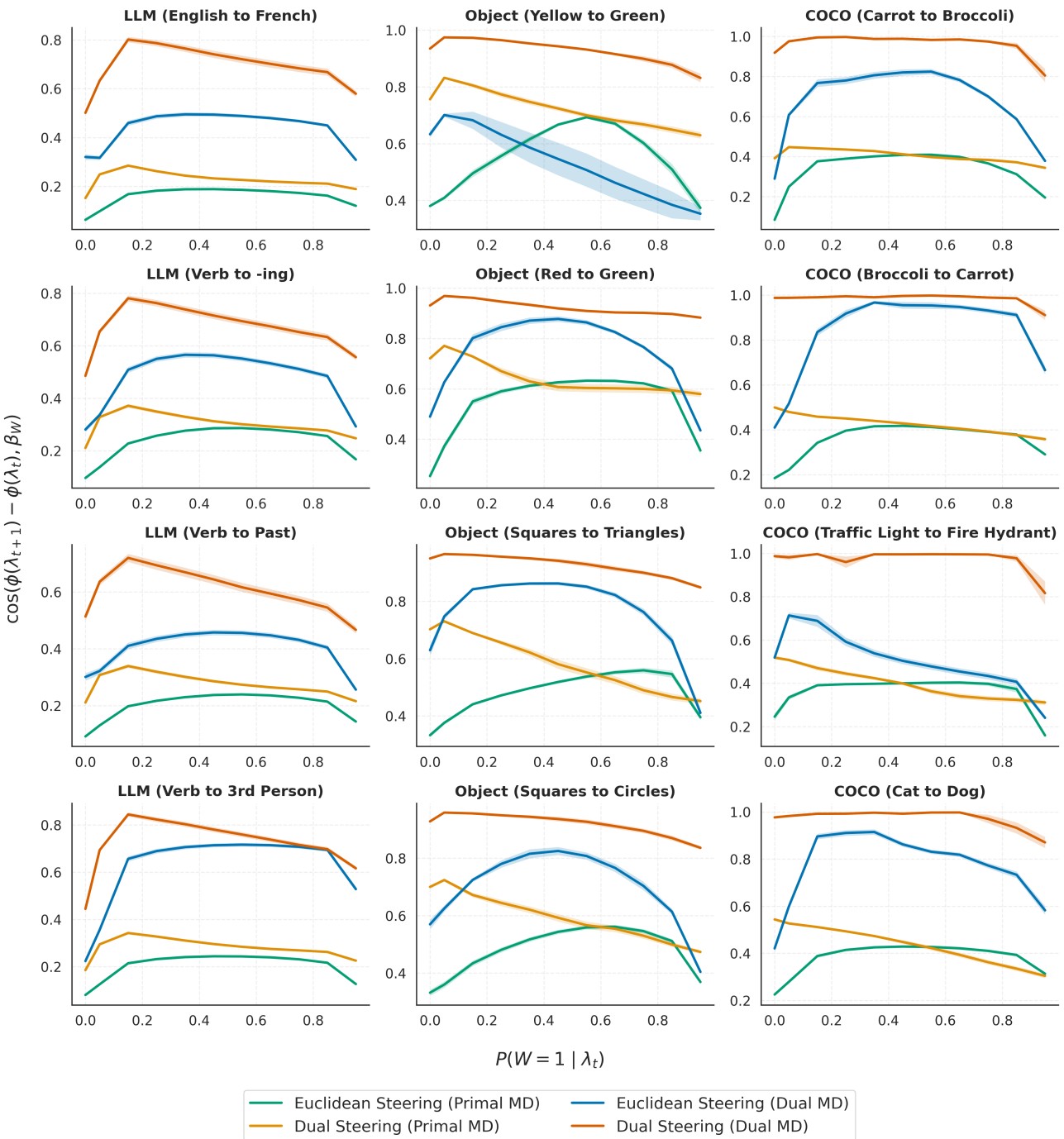

*Figure 6.* Cosine similarity between the concept direction and the change in dual coordinates during Euclidean (green and blue) and dual steering (orange and red). Dual steering maintains a higher cosine similarity than Euclidean steering, indicating that dual steering effectively moves along the concept direction in the dual space. In particular, the increase in cosine similarity during the first few steps suggests that the regularized Newton method initially adjusts the trajectory to increase the local variance of the concept, facilitating effective dual steering along the concept direction. Lines represent the mean, and shading indicates the standard error of the mean (SEM) across test contexts.

## C.2. Primal and Dual MDs as Linear Probes

Figure 7 illustrates the effectiveness of primal and dual mean differences as linear probes for the target concepts across all experiments. Both methods successfully separate the base and target context embeddings in the test sets, demonstrating their

utility in probing for the respective concepts.

However, Figure 8 evaluates the validity of the probing assumption in Theorem 3.1. For each Euclidean or dual steering path $\{\lambda_t\}$, the figure displays both $\operatorname{logit} P(W = 1 \mid \lambda_t)$ and the projection onto the concept direction $\beta_W^\top \lambda_t / \|\beta_W\|_2$. When compared to the logits and projections of the context embeddings in the test set, we observe that dual steering paths typically yield lower logit values than those of the test contexts at the same projection value. This suggests that the target concept probability $P_\lambda^W(1)$ is not constant along the hyperplane defined by the linear probe. Consequently, the probing assumption is not strictly satisfied in practice; as discussed in Section 5.3, this discrepancy may impact the overall effectiveness of dual steering.

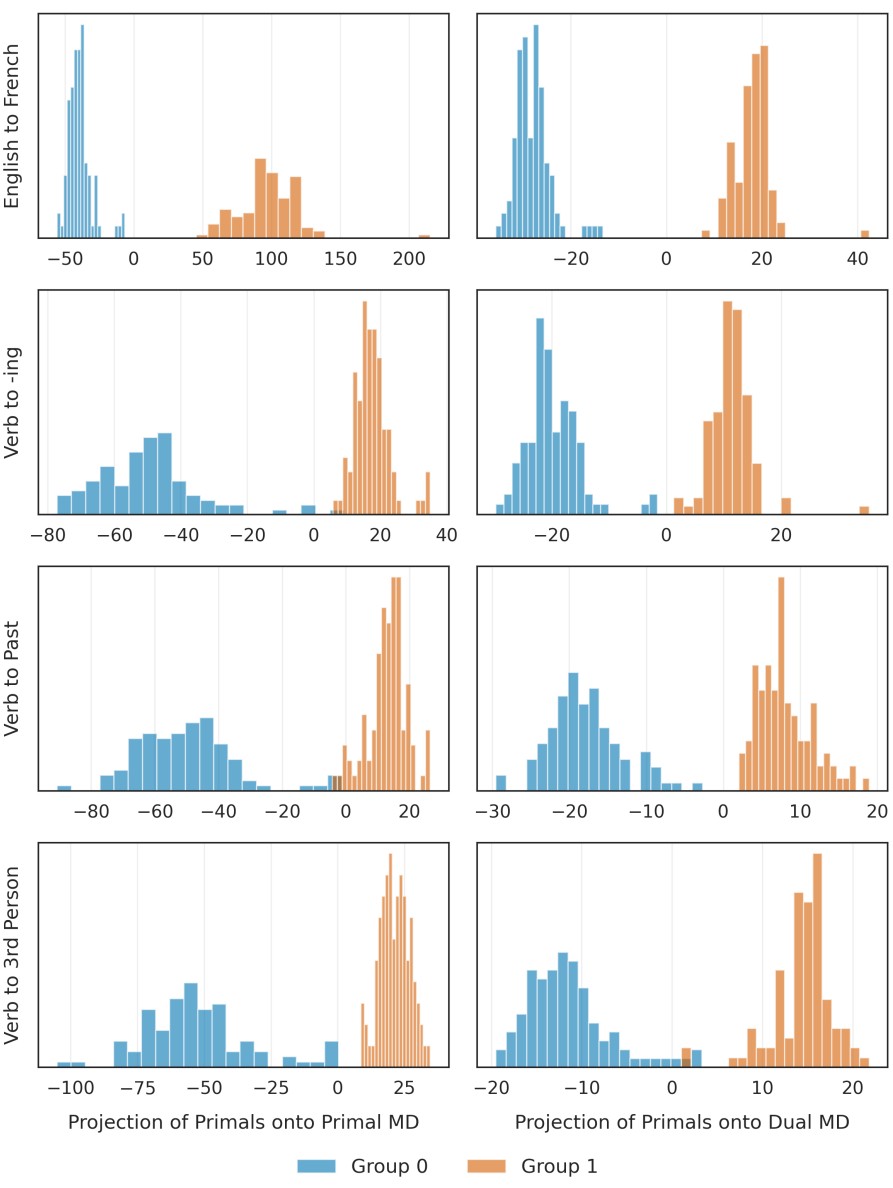

*Figure 7.* Histogram of projections of test set context embeddings onto primal and dual mean differences $\beta_W^\top \lambda_i / \|\beta_W\|_2$. Both primal and dual mean differences effectively separate the base and target context embeddings, indicating their utility as linear probes for the respective concepts.

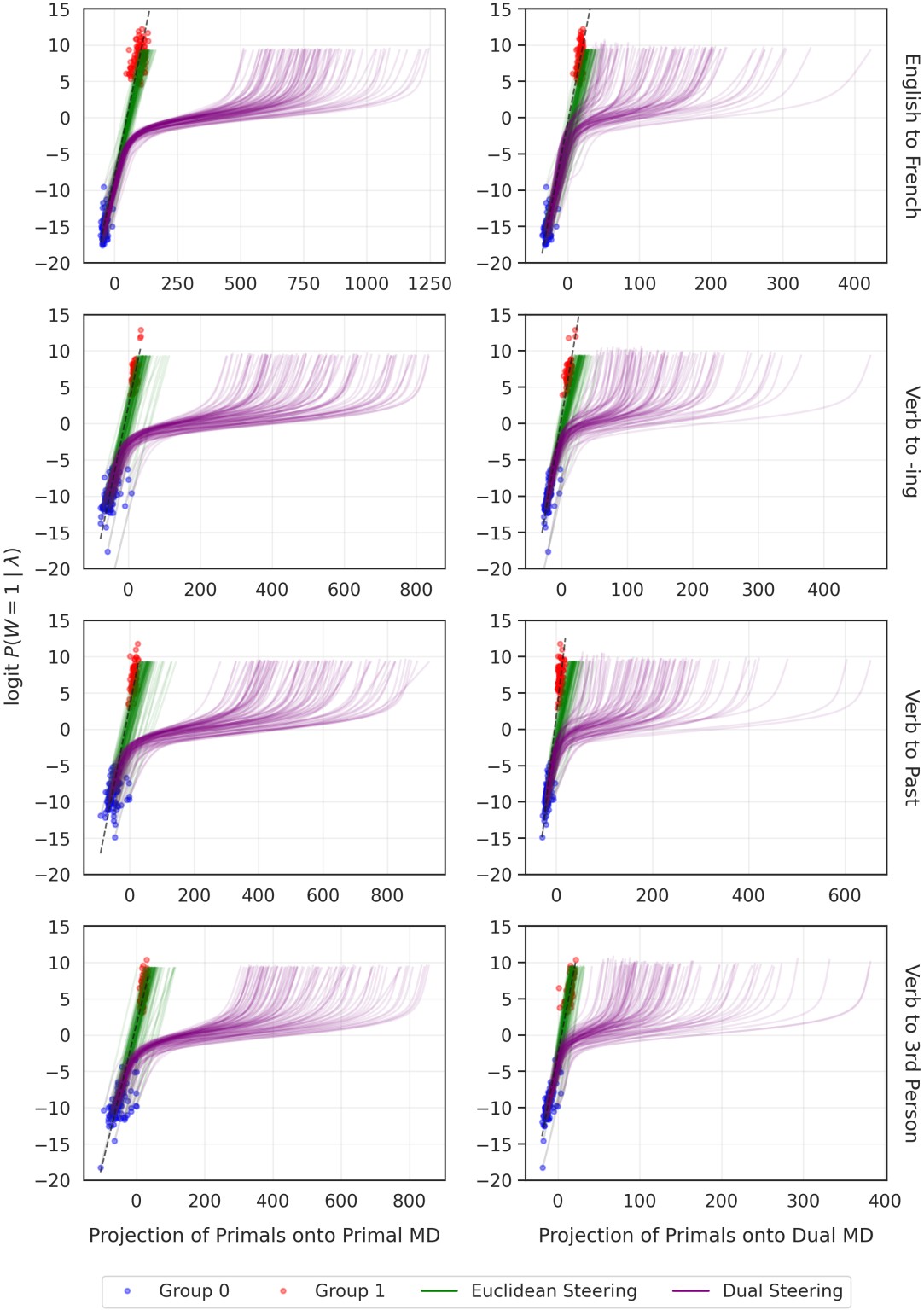

*Figure 8.* Projection of Euclidean (green) and dual (purple) steering paths onto the linear probe ($\beta_W^\top \lambda_t / \|\beta_W\|_2$) against the logit of the target concept probability ($\mathrm{logit}\, P(W = 1 \mid \lambda_t)$), using primal or dual mean differences as the linear probe. Blue and red dots represent the projections and logits of context embeddings from the base and target test groups, respectively. Dual steering typically yields a lower concept probability for any given projection value compared to the test set contexts. This suggests that the probing assumption in Theorem 3.1 is not strictly satisfied in practice.

## C.3. Steering Results for More Concepts and Directions

In this section, we present additional steering results for additional concepts with both the LLM and CLIP experiments as well as additional steering directions.

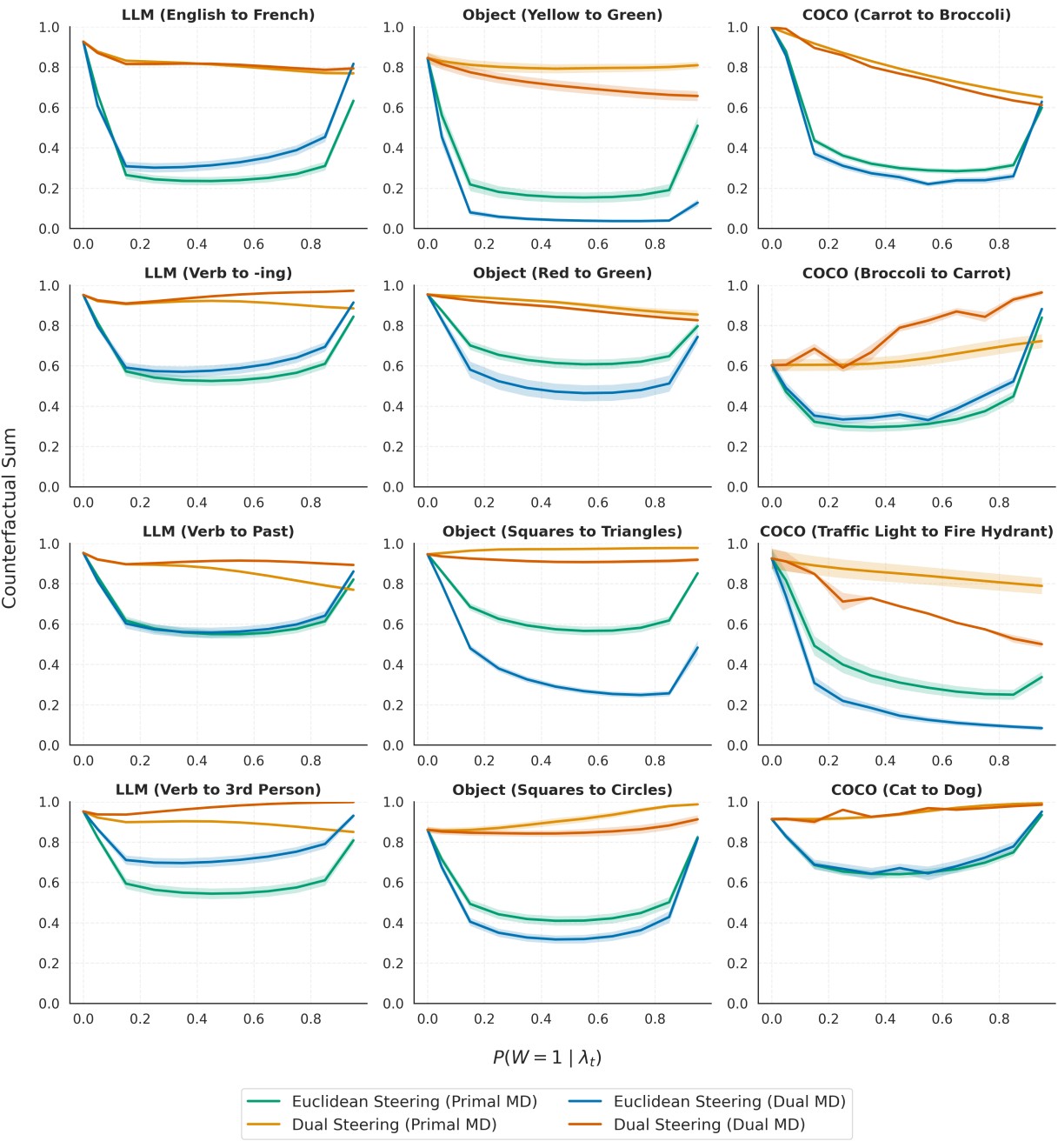

*Figure 9.* Total probability mass on counterfactual pairs during Euclidean and dual steering across all experiments. Dual steering consistently preserves a higher probability mass on counterfactual pairs during intermediate steps; this suggests greater robustness, as it avoids "leakage" to neutral tokens more effectively than Euclidean steering. Lines represent the mean, and shading indicates the standard error of the mean (SEM) across test contexts.

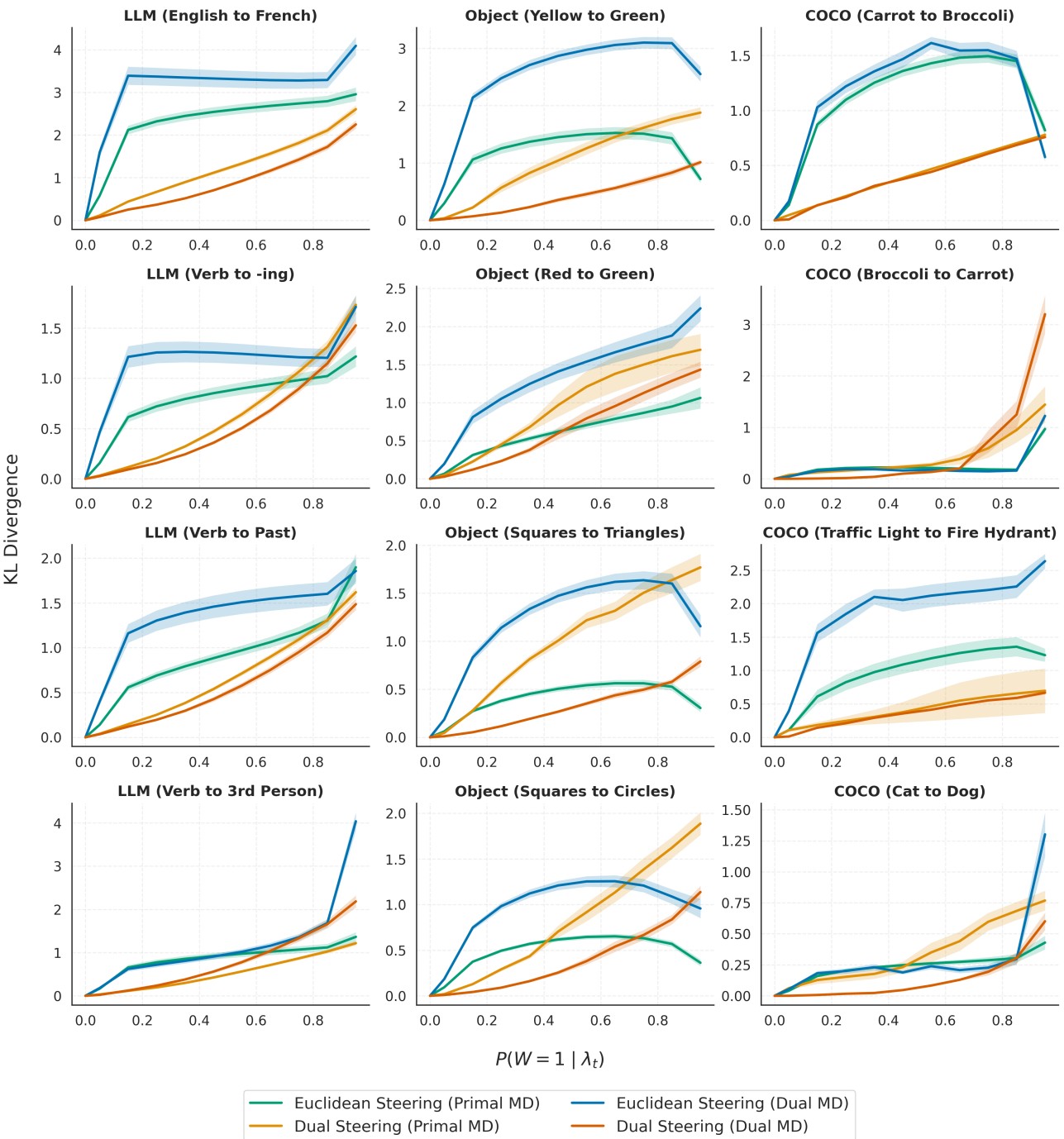

*Figure 10.* KL divergence of off-target distributions during Euclidean and dual steering across all experiments. Dual steering results in lower KL divergence values, indicating better preservation of off-target concepts compared to Euclidean steering. Lines represent the mean, and shading indicates the standard error of the mean (SEM) across test contexts.

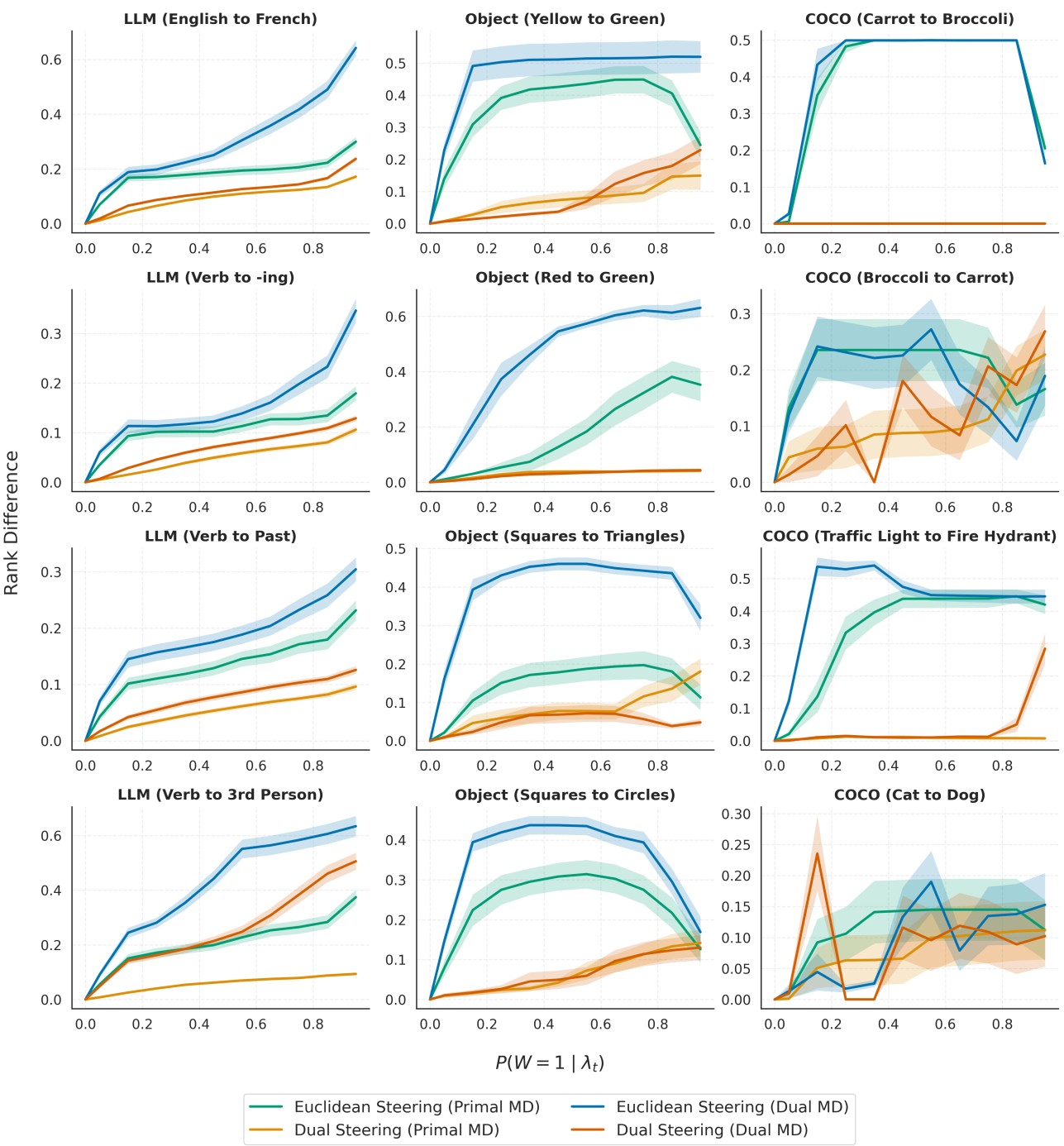

*Figure 11.* Rank differences of off-target distributions during Euclidean and dual steering across all experiments. Dual steering results in lower rank differences, indicating better preservation of off-target concepts compared to Euclidean steering. Lines represent the mean, and shading indicates the standard error of the mean (SEM) across test contexts.

