# OpenReview forum: "The Information Geometry of Softmax: Probing and Steering"
_ICML.cc/2026/Conference — ICML 2026 regular_

### Official Review · Reviewer_JieR · 2026-03-07

**Soundness:** 3
**Presentation:** 3
**Significance:** 3
**Originality:** 4
**Overall Recommendation:** 4
**Confidence:** 3

**Summary:**

This paper studies the geometric structure of softmax-parameterized representation spaces and how this structure affects concept probing and steering. The authors argue that the natural geometry of such representations is not Euclidean, but information geometry; under this framework, representations admit both primal and dual structures, which correspond to different forms of interpolation and manipulation. Based on this observation, the paper proposes dual steering, and supports both theoretically and empirically that this method preserves off-target distributions better than standard Euclidean steering. Experiments are conducted on Gemma-3-4B and MetaCLIP-2.

**Compliance With Llm Reviewing Policy:**

Affirmed.

**Final Justification:**

The authors' rebuttal has addressed my main concerns, and I recommend acceptance of this work.

**Key Questions For Authors:**

1. How do the conclusions extend to intermediate-layer steering? Why not include at least a preliminary experiment?
2. What are the computational costs of dual steering compared to Euclidean steering, in terms of runtime, memory, and scalability?
3. Have the authors evaluated the method on more models or tasks? What are the most promising application scenarios?
4. How does dual steering degrade when the concept-factorizable assumption does not hold?

**Limitations:**

Yes.

**Strengths And Weaknesses:**

**Strengths:**
- The paper has clear motivation, a unified narrative, and strong originality. Starting from the insight that softmax representations ultimately define output distributions, so their geometry should align with distribution differences, the authors connect KL/Bregman geometry, primal/dual structure, interpolation semantics, and steering methods into a coherent story.
- Connecting information geometry with the common linear representation/steering perspective and deriving dual steering is one of the paper’s most valuable contributions.
- Although the experiments are limited in scope, they consistently support the claim that dual steering induces less off-target drift than Euclidean steering.


**Weaknesses:**
- The main concern is significance and external validity. This paper studies probing/steering geometry in softmax output parameter space, while more common and impactful interventions typically occur in intermediate layers. The authors acknowledge that extending this geometric understanding to intermediate layers remains future work. This makes the paper feel more like an elegant theoretical perspective in a clean setting, but with a noticeable gap from broader practice.
- The engineering usability of the method is unclear. Dual steering requires repeatedly estimating covariance, adding regularization terms, solving linear systems, and iterative updates; it is not a simple one-line replacement for Euclidean steering. The paper provides algorithmic and geometric explanations but lacks runtime, memory, step sensitivity, or scalability analysis.
- The experiments remain largely at the theoretical validation level. Evaluations are only on Gemma-3-4B and MetaCLIP-2, covering both text and visual concepts, but still read more like proof-of-concept. The theoretical results also rely on strong concept-factorizable and ideal probe assumptions, which the paper acknowledges do not always hold in practice.

---

> ### Author Rebuttal · Authors · 2026-03-31
>
> Thank you for your constructive review!
>
> **(1) Intermediate layers**
>
> The core idea of this paper is that closeness of activations should reflect closeness of the induced output distributions. The “correct” geometry at an intermediate layer would, in principle, be defined by pushing the activation through all remaining components of the model. For more complex concepts, this would involve aggregating KL divergences over all subsequent positions in which the activation participates within the context window. While this formulation is conceptually sound, it is analytically intractable in practice. Nevertheless, our results in the one tractable setting (the pure softmax case) demonstrate that this distinction between activation-space closeness and distributional closeness is indeed important.
>
> A promising direction for intermediate layers is to formulate a learning problem for the geometry itself. Specifically, one could parametrize a distance function using a neural network and train it to approximate the idealized aggregated output KL divergence. This may provide a tractable approximation to the true geometry and represents an interesting direction for future work.
>
> **(2) Computational cost of dual steering**
>
> Dual steering involves computing the Fisher information matrix and its inverse, which has a high computational complexity $O(kd^2 + d^3)$ where $k$ is the vocabulary size (e.g., 262K for LLMs) and $d$ is the hidden dimension (e.g., 2K for LLMs). To address this, we approximated the covariance matrix using only the Top-K (e.g., 20,000) tokens at each step, which means $k$ becomes much smaller. This approximation remains highly accurate because the vast majority of the probability mass is concentrated on a few thousand tokens.
>
> While this makes the algorithm much faster, we have developed a new algorithm to improve the computation cost. Rather than computing the covariance matrix and inverse separately, we instead solve directly the linear system $(\mathrm{Cov}[\gamma \mid \lambda] + \alpha I_d)v = \epsilon \beta_W$ in Section 4.2 using a conjugate gradient method.
> Then, the computation complexity becomes $O(nkd)$ where $n$ is the number of conjugate gradient iterations (e.g., $n$ = 20). Also, we have checked that this method is much faster than our original algorithm, while providing the same results. We will add a description about this new algorithm and all the results will be from this algorithm. Thank you for the comments on the computation costs!
>
> **(3) Experiments on other models or tasks**
>
> It is easy to apply the code (we will release it on GitHub) to any other models by changing the model name and the vocabulary format. Since our results are theoretically grounded, we expect similar qualitative behavior to hold across different models. For tasks beyond binary concepts, as long as there is a specific target hyperplane, we can apply dual steering to find the KL minimizer. However, it is not obvious what the off-target structure should be. Moreover, for common steering applications, like sycophancy, toxicity, or hallucinations, we would need to evaluate full generations (rather than a single next token distribution).
>
> **(4) Concept factorizability**
>
> Concept factorization is an idealization that we suspect is likely stronger than what is really required in practice. However, we like it because it’s both conceptually clear and allows for a compact argument that we feel exposes the main ideas relatively clearly.
>
> We need this assumption for the proof of Theorem 3.2 (especially to derive eq. A.26 from eq. A.25). If we try to state a weakened version of the assumption without factorization, we would take the same off-target distribution $P^Z$ and define $P^{W_i}$ for each counterfactual pair. Then, to provide a rigorous proof, we need to handle the variation in the KL of $P^{W_i}$, which makes the first term in eq. A.25 non-constant in the optimization problem. This is a substantial increase in complexity.
>
> Note, however, that even in this case, because the counterfactual sum is still constant for dual steering, dual steering would outperform Euclidean steering. Indeed, the mixture structure (off-target distribution) between counterfactual pairs and the neutral tokens is the main point, so factorizability itself is not that important when compared to Euclidean steering. For example, when you see the bottom left plot of Figure 1, for the same x-axis value, P(“operates”) / P(“operate”) is not the same as P(“maintains”) / P(“maintain”), but overall off-target distribution is preserved.

---

> > ### Author Rebuttal · Reviewer_JieR · 2026-04-04
> >
> > The new conjugate gradient algorithm significantly improves the practical value of this work. My main concerns are addressed, and will adjust my score accordingly.

---

> > > ### Author Response · Authors · 2026-04-04
> > >
> > > Thank you for your support. We will update the algorithm in the revised version of the paper.

---

### Official Review · Reviewer_2AUg · 2026-03-08

**Soundness:** 3
**Presentation:** 3
**Significance:** 3
**Originality:** 3
**Overall Recommendation:** 4
**Confidence:** 3

**Summary:**

This paper proposes a re-examination of the hidden layer space from the perspective of information geometry, assuming that current research conventions default to treating it as a flat Euclidean space. The authors argue that since the Transformer output layer (Softmax) ultimately outputs a probability distribution, the corresponding logit space can be interpreted as the "natural parameter space" of an exponential family distribution.

Based on this modeling, the paper points out that traditional Euclidean steering suffers from geometric inconsistencies, ignoring the curvature structure of the space. This can lead to "conceptual leakage" or unnecessary distribution shifts when modifying the target semantics. To correct this geometric inconsistency, the authors propose two alternatives: natural steering and dual steering.

**Compliance With Llm Reviewing Policy:**

Affirmed.

**Final Justification:**

My concerns have been fully addressed.

**Key Questions For Authors:**

(1) In the high-dimensional space of large language models, the precise calculation of the Fisher Information Matrix obviously incurs high computational costs. Are there any efficient approximation schemes? More importantly, will such approximation strategies compromise geometric fidelity?

(2) Natural Steering is essentially a linearized approximation of local manifold geometry. If the steering amplitude is large, does this linear assumption still hold? Furthermore, in this case, will the cumulative effect of curvature lead to KL divergence explosion, thus causing model output instability?

(3) In practical applications, the application boundaries of the two proposed steering strategies are rather vague. Are there clear theoretical or experimental standards to distinguish their applicable scenarios? Or, on which specific downstream tasks will Dual Steering perform better than Natural Steering (and vice versa)?

**Limitations:**

This paper discusses some theoretical and practical limitations but could further elaborate on the sensitivity to noise in Fisher information matrix estimation.

**Strengths And Weaknesses:**

Strengths:

(1) This paper does not treat representation steering as a simple vector addition or subtraction. Instead, it takes a geometric approach, modeling the Transformer output layer as a natural parameter space with an exponential family distribution, thus maintaining geometric consistency.

(2) The authors astutely recognize a common assumption in current research: that the latent space is a flat Euclidean space. They point out a potential geometric inconsistency behind this linear operation, which has significant theoretical reconstruction value.

(3) The experimental section of the paper maintains good consistency with theoretical claims. It not only focuses on the performance of downstream tasks but also delves into the behavior of KL divergence, the stability of non-target distributions, and curvature-related properties.

Weaknesses:

(1) The core of Natural Steering lies in the estimation of the Fisher Information Matrix. However, in the high-dimensional representation space of LLMs, the cost of accurately computing or storing this matrix is extremely high. The paper, however, offers limited discussion of scalable approximation methods.

(2) The experiments in the paper primarily focus on controlled semantic properties. While this is effective for validating geometric effects, the effectiveness of the method in more complex real-world alignment tasks (such as alleviating hallucinations) remains unclear.

(3) Natural Steering is inherently based on local properties. If the steering force increases, will the cumulative effect of manifold curvature render the local approximation ineffective? Further exploration of this boundary will contribute to the theoretical completeness of the framework

---

> ### Author Rebuttal · Authors · 2026-03-31
>
> Thank you for the comprehensive review. You mentioned “the authors propose two alternatives: natural steering and dual steering” in the summary and mentioned some points on “Natural Steering”. However, we propose only one alternative to Euclidean steering: dual steering. (There is no “natural steering”). Since we are a bit unsure about what you mean by “natural steering,” we cannot respond to all the questions perfectly. But we want to respond to some points on “computational cost of dual steering”, “local approximation”, and “extension to more complex concepts”.
>
> **(1) Computational cost of dual steering**
>
> Dual steering involves computing the Fisher information matrix and its inverse, which has a high computation complexity $O(kd^2 + d^3)$ where $k$ is the vocabulary size (e.g., 262K for LLMs) and $d$ is the hidden dimension (e.g., 2K for LLMs). To address this, we approximated the covariance matrix using only the Top-K (e.g., 20,000) tokens at each step, which means $k$ becomes much smaller. This approximation remains highly accurate because the vast majority of the probability mass is concentrated on a few thousand tokens.
>
> While this makes the algorithm much faster, we have developed a new algorithm to improve the computation cost. Rather than computing the covariance matrix and inverse separately, we instead directly solve the linear system $(\mathrm{Cov}[\gamma \mid \lambda] + \alpha I_d)v = \epsilon \beta_W$ in Section 4.2 using a conjugate gradient method.
> Then, the computational complexity becomes $O(nkd)$ where $n$ is the number of conjugate gradient iterations (e.g., $n$ = 20). Also, we have checked that this method is much faster than our original algorithm while providing the same results. We will add a description about this new algorithm and all the results will be from this algorithm. Thank you for the comments on the computation costs!
>
> **(2) Local approximation**
>
> If “natural steering” refers to Euclidean steering, then of course, the local approximation around the starting point totally fails as the steering magnitude becomes larger. If “natural steering” refers to dual steering, a sufficiently small steering step size gives a good enough approximation between the small steps in the primal and dual spaces. Moreover, due to the feasibility issue (dual coordinates should be in the convex hull of unembedding vectors), we usually cannot get the dual step perfectly parallel to the given linear probe. Thus, the gap of the local approximation for each small step is not significant to the effectiveness of the dual steering method.
>
> **(3) More complex concepts**
>
> We would need to see long generated sequences in order to check the more complex concept (e.g. hallucinations), and thus we would likely need to steer in intermediate layers. However, the core idea of this paper is that closeness of activations should reflect closeness of the induced output distributions. The “correct” geometry at an intermediate layer would, in principle, be defined by pushing the activation through all remaining components of the model. For more complex concepts, this would involve aggregating KL divergences over all subsequent positions in which the activation participates within the context window. While this formulation is conceptually sound, it is analytically intractable in practice. Nevertheless, our results in the one tractable setting (the pure softmax case) demonstrate that this distinction between activation-space closeness and distributional closeness is indeed important.
>
> A promising direction for intermediate layers is to formulate a learning problem for the geometry itself. Specifically, one could parametrize a distance function using a neural network and train it to approximate the idealized aggregated output KL divergence. This may provide a tractable approximation to the true geometry and represents an interesting direction for future work.

---

> > ### Author Rebuttal · Reviewer_2AUg · 2026-04-03
> >
> > Thanks to the author for the clarification regarding the steering terminology and the additional details on computational optimization (Top-K truncation and conjugate gradient method). However, I still have the following concerns:
> > 1. The rebuttal mentions that “This approximation remains highly accurate,” but it does not provide any quantitative error analysis for approximation.
> > 2. The robustness under large steering magnitudes remains unclear.

---

> > > ### Author Response · Authors · 2026-04-04
> > >
> > > Thanks for your follow-up.
> > >
> > > 1. For the approximation of the Fisher information matrix, we can compare our top-k approximation to the full Fisher matrix using the Frobenius norm of the difference between the two matrices along the steering trajectory. Below is a representative example. You can see that the difference is near zero at top-20k:
> > >
> > > | TopK | Step 0 | Step 5 | Step 10 | Step 50 | Step 100 |
> > > | --- | --- | --- | --- | --- | --- |
> > > | 1 | 0.2031 | 0.2031 | 0.2031 | 0.2926 | 0.3321 |
> > > | 20 | 0.0120 | 0.0133 | 0.0151 | 0.0006 | 0.0000 |
> > > | 2000 | 0.0001 | 0.0002 | 0.0002 | 0.0000 | 0.0000 |
> > > | 20000 | 0.0000 | 0.0000 | 0.0000 | 0.0000 | 0.0000 |
> > > | 200000 | 0.0000 | 0.0000 | 0.0000 | 0.0000 | 0.0000 |
> > >
> > > 2. With respect to robustness of the steering method: Let us first emphasize that the method only relies on a *locally* linear approximation to the geometry. To steer using a dual steering vector with significant magnitude, we take many local steps; the dual steering procedure involves setting a step size and a step count. All of the experiments reporting in the paper follow this procedure. That is, the reported robustness results are all in the "large steering magnitude" regime.

---

### Official Review · Reviewer_QW1U · 2026-03-13

**Soundness:** 3
**Presentation:** 3
**Significance:** 3
**Originality:** 4
**Overall Recommendation:** 5
**Confidence:** 4

**Summary:**

This paper argues that the natural geometry of softmax representation spaces is the Bregman (dually flat) geometry induced by the KL divergence, not the Euclidean geometry typically assumed. A central concept examined by the paper is the duality structure this geometry induces, which reveals two semantically distinct interpolation modes and exposes a "type error" in standard representation steering. This paper assesses a major issue in current steering methods: adding a probe vector directly in the primal space causes off-target drift. The authors propose dual steering, which operates in the dual coordinate system, and prove it optimally preserves off-target distributions under a concept-factorizability assumption. Experiments on Gemma-3-4B and MetaCLIP-2 confirm consistent improvements over Euclidean steering across multiple robustness metrics.

**Compliance With Llm Reviewing Policy:**

Affirmed.

**Final Justification:**

I am satisfied with the author response.

**Key Questions For Authors:**

I don't have questions.

**Limitations:**

yes

**Strengths And Weaknesses:**

The paper's core theoretical contribution is clean and compelling. The observation that standard steering commits a "type error" by adding a dual-space object in the primal space is simple yet important, and Theorem 3.2 provides a rigorous optimality guarantee for the proposed fix. The duality between AND and OR interpolation semantics is elegant and effectively demonstrated across both LLMs and CLIP. The experimental evaluation is reasonably thorough, spanning two modalities, multiple concepts, and three complementary off-target metrics that all tell a consistent story. Overall, this is an important contribution to the literature on linear representations in LLMs and using them to control LLM behavior.

The concept-factorizability assumption (Definition 3.1) is pretty strong and unlikely to hold exactly in real settings, yet the paper does not quantify how gracefully performance degrades as this assumption is violated. Relatedly, the probing assumption that the concept probability is constant across entire hyperplanes is empirically violated as presented in the figure 6, which undermines the formal guarantees that motivate the method. Perhaps most importantly, all experiments operate at the final softmax layer, whereas practitioners typically steer at intermediate layers, leaving a significant gap between the theory and real-world applicability that is only briefly acknowledged as future work. Finally, the paper only compares against naive Euclidean steering and does not benchmark against more sophisticated baselines such as activation engineering variants or the causal inner product methods of Park et al or the work [5].

The related work discussion and the overall framing overlooks a substantial body of prior work on concept erasure and the linear representation hypothesis that predates the contributions of Park et al., including [1], [2], [3], [4], and [5]. This omission is notable because several of these works are directly relevant to the paper's core contributions. In particular, [5] already frames steering as finding a minimal perturbation under some distance metric subject to the constraint of moving along the concept subspace, which is essentially the same optimization perspective presented in Section 3.1. This work should be cited and discussed in that section, with a clear explanation of how the proposed approach differs from or builds upon it.

[1] Bolukbasi, Tolga, et al. "Man is to computer programmer as woman is to homemaker? debiasing word embeddings." Advances in neural information processing systems 29 (2016).


[2] Vargas, Francisco, and Ryan Cotterell. "Exploring the linear subspace hypothesis in gender bias mitigation." Proceedings of the 2020 Conference on Empirical Methods in Natural Language Processing (EMNLP). 2020.


[3] Belrose, Nora, et al. "Leace: Perfect linear concept erasure in closed form." Advances in Neural Information Processing Systems 36 (2023): 66044-66063.

[4] Ravfogel, Shauli, et al. "Linear adversarial concept erasure." International Conference on Machine Learning. PMLR, 2022.

[5] Singh, Shashwat, et al. "Representation surgery: theory and practice of affine steering." Proceedings of the 41st International Conference on Machine Learning. 2024.

---

> ### Author Rebuttal · Authors · 2026-03-31
>
> Thanks for your thoughtful review and your engagement with our work!
>
> **(1) Concept factorizability**
>
> Concept factorization is an idealization that we suspect is likely stronger than what is really required in practice. However, we like it because it’s both conceptually clear and allows for a compact argument that we feel exposes the main ideas relatively clearly.
>
> We need this assumption for the proof of Theorem 3.2 (especially to derive eq. A.26 from eq. A.25). If we try to state a weakened version of the assumption without factorization, we would take the same off-target distribution $P^Z$ and define $P^{W_i}$ for each counterfactual pair. Then, to provide a rigorous proof, we need to handle the variation in the KL of $P^{W_i}$, which makes the first term in eq. A.25 non-constant in the optimization problem. This is a substantial increase in complexity.
>
> Note, however, that even in this case, because the counterfactual sum is still constant for dual steering, dual steering would outperform Euclidean steering. Indeed, the mixture structure (off-target distribution) between counterfactual pairs and the neutral tokens is the main point, so factorizability itself is not that important when compared to Euclidean steering. For example, when you see the bottom left plot of Figure 1, for the same x-axis value, P(“operates”) / P(“operate”) is not the same as P(“maintains”) / P(“maintain”), but overall off-target distribution is preserved.
>
> **(2) Probing assumption**
>
> Extending the analysis to settings with probe misspecification is an important direction for future work. Based on the empirical results, we do suspect that the dual steering is more robust to probe misspecification. However, it’s not obvious what the right way to think of this misspecification is.
>
> **(3) Intermediate layers**
>
> The core idea of this paper is that closeness of activations should reflect closeness of the induced output distributions. The “correct” geometry at an intermediate layer would, in principle, be defined by pushing the activation through all remaining components of the model. For more complex concepts, this would involve aggregating KL divergences over all subsequent positions in which the activation participates within the context window. While this formulation is conceptually sound, it is analytically intractable in practice. Nevertheless, our results in the one tractable setting (the pure softmax case) demonstrate that this distinction between activation-space closeness and distributional closeness is indeed important.
>
> A promising direction for intermediate layers is to formulate a learning problem for the geometry itself. Specifically, one could parametrize a distance function using a neural network and train it to approximate the idealized aggregated output KL divergence. This may provide a tractable approximation to the true geometry and represents an interesting direction for future work.
>
> **(4) Benchmarking**
>
> This is a good point. The main idea of dual steering is that it is theoretically principled, as it reflects the real probability distribution, and allows us to move beyond linear steering in the primal space. We compare it with Euclidean steering, which is a representative method that neither reflects the probability distribution nor departs from linear operation in the primal space. While comparisons with other variants are also reasonable, we expect similar results, such as the leakage phenomenon. We will include these comparisons in the revised version of the paper.
>
> **(5) Related work**
>
> Thank you for the suggestions on the related work. We will include a discussion of the connection to the concept erasure literature in an updated version of the paper. We also appreciate the suggestion of Singh et al., as this work is very related. Similar to our approach, they formulate steering as a minimization problem. However, they rely on affine transformations of representations and L2 distance, whereas we use KL divergence as a more principled notion of distance. Regarding concept erasure, one can view it as a projection onto the hyperplane corresponding to the target concept probability $P^W(1) = 0.5$. In this context, dual steering identifies the point that best preserves the off-target distribution while enforcing the erasure of the target concept.

---

> > ### Author Rebuttal · Reviewer_QW1U · 2026-04-03
> >
> > Thank you for your for your response. My concerns have been resolved and I will increase my score to "accept". I hope the RW will be updated.

---

> > > ### Author Response · Authors · 2026-04-04
> > >
> > > Thank you, and thanks for your support of our work. We will update the related work in the revised version of the paper.

---

### Official Review · Reviewer_C4hX · 2026-03-20

**Soundness:** 3
**Presentation:** 3
**Significance:** 3
**Originality:** 3
**Overall Recommendation:** 5
**Confidence:** 4

**Summary:**

The author's aim to characterize the latent geometry of AI models whose outputs are softmax distributions defined by representation vectors $\lambda$ (an exponential family with natural parameter $\lambda$). In particular, the author's first introduce a Bregman geometry induced by the KL-divergence between softmax distributions with different representation vectors; here, they are able to define the dual map of a representation vector $\phi(\lambda) := \mathbb{E}[\gamma\mid \lambda]$, and by means of the convex conjugate, a bijection between the representation vector $\lambda$ (primal coordinates) and its dual map $\phi(\lambda)$ (dual coordinates).

With this Bregman geometry in hand, the authors go on to study how the output distribution changes while interpolating between either primal or dual coordinates. Their findings and theory (Proposition 2.1) show that minimizing reverse-KL is equivalent to primal interpolation and further, primal interpolations acts like an AND operation applied to high probability regions of the two distributions being interpolated. In contrast, minimizing forward-KL is equivalent to dual interpolation which takes the union (OR operation) of high-probability regions between the two distributions.

Based on this interpolation result, the authors propose a dual steering (wrt their dual map) method which they contrast with the largely popular Euclidean steering. Under a factorization assumption on the output distribution, the authors show that their dual steering method preserves off-target concepts while only modifying the desired on-target concept which differs from Euclidean steering in that Euclidean steering may still change off-target concepts under this assumption. The authors then provide a practical algorithm for performing dual steering via a Newton-based update.

In the authors' experiments, they contrast the performance of Dual vs Euclidean steering applied to an Gemma-3-4B and MetaClip-2 using 3 robustness metrics. Their experiments show that, with respect to their metrics, Dual steering maintains probability mass on counterfactual pairs (outputs encoding the targeted concept) and preserves the off-target distribution better than Euclidean steering.

**Compliance With Llm Reviewing Policy:**

Affirmed.

**Final Justification:**

The authors have thoroughly addressed my main concerns which leads me to recommend acceptance of this strong work.

**Key Questions For Authors:**

Can the assumption on Concept-Factorizable Distributions be weakened in Theorem 3.2 to achieve a weakened result similar to eqn (3.3)? In practice, this assumption does not often seem to be satisfied (e.g. the authors' experimental results); Maybe this isn't possible considering some of the results in Figure 9? It would also be interesting to see where this result breaks down.

Can these results be extended outside of binary concepts?

Can we obtain results in the setting where we assume a linear probe with error, i.e., can we robustify the results wrt the probe that we have in practice (Figure 6)?

**Limitations:**

Yes

**Strengths And Weaknesses:**

The theoretical claims made in the paper (proposition 2.1 and Theorem 3.2) are sound and the proofs are correct (albeit a typo in eqn (A.27) in the second sum, though the result still holds). The assumption of access to a "correct" linear probe is strong in practice, but seems to facilitate the analysis. Experiments are performed for each relevant result; in particular, these experiments are performed on two different modalities: text and image, which is great. With respect to their Theorem 3.2, the results in Section 5 provide strong empirical justification for their proposed dual steering method.

The work is well-structured with coherent organization of the content. I think section 2 could be better connected to section 3; in particular, I think proposition 2.1 should be connected to Theorem 3.2 since dual steering seems like it can be thought as a further generalization of dual interpolation (extending from two points to an affine subspace).

Steering is a very popular problem with many avenues of improvement, especially in terms of theory. The results provided here give a principled verification of the phenomenon of off-target drift and argue for a dual steering method based upon the information geometry induced by softmax distributions (exponential families). Furthermore, the authors provide a novel theoretical framework for analyzing steering via the perspective of information geometry that is supported by their empirical results across both text and image modalities.

---

> ### Author Rebuttal · Authors · 2026-03-31
>
> Thank you for your careful review and for your support!
>
> **(1) Can the concept factorization assumption be weakened?**
>
> Concept factorization is an idealization that we suspect is likely stronger than what is really required in practice. However, we like it because it’s both conceptually clear and allows for a compact argument that we feel exposes the main ideas relatively clearly.
>
> We need this assumption for the proof of Theorem 3.2 (especially to derive eq. A.26 from eq. A.25). If we try to state a weakened version of the assumption without factorization, we would take the same off-target distribution $P^Z$ and define $P^{W_i}$ for each counterfactual pair. Then, to provide a rigorous proof, we need to handle the variation in the KL of $P^{W_i}$, which makes the first term in eq. A.25 non-constant in the optimization problem. This is a substantial increase in complexity.
>
> Note, however, that even in this case, because the counterfactual sum is still constant for dual steering, dual steering would outperform Euclidean steering. Indeed, the mixture structure (off-target distribution) between counterfactual pairs and the neutral tokens is the main point, so factorizability itself is not that important when compared to Euclidean steering. For example, when you see the bottom left plot of Figure 1, for the same x-axis value, P(“operates”) / P(“operate”) is not the same as P(“maintains”) / P(“maintain”), but overall off-target distribution is preserved.
>
> **(2) Can the results be extended outside of binary concepts?**
>
> A key challenge is making sense of what linear representation means in the non-binary case. For categorical concepts, we can reduce the problem to binary concepts similar to Park et al., “The Geometry of Categorical and Hierarchical Concepts in Large Language Models”. For continuous scalar variables, if we partition the vocabulary into related tokens and neutral tokens, then we would expect Euclidean steering to still exhibit “leakage” in the same manner. For example, a similar result holds if we assume that the linear probe satisfies the condition that the proportions of probability mass assigned to each related token are identical on the hyperplane it defines.
>
> **(3) Can we obtain results in the setting where we assume a linear probe with error?**
>
> This is a great direction for future work. Based on the empirical results, we do suspect that the dual steering is more robust to probe misspecification. However, it’s not obvious what the right way to think of this misspecification is.
>
> **(4) Other points:**
> - Thanks for pointing out the typo in eq. A.27. We will fix this in an updated version of the paper.
> - Thanks for the suggestion about clarifying the relationship between Section 2 (interpolation) and Section 3 (steering). We will add a description of the connection between interpolation and steering. Dual steering preserving the off-target distribution can be connected to the behavior of dual interpolation that takes a linear mixture of the endpoints, while the leakage phenomenon of Euclidean steering can be connected to the behavior of primal interpolation that takes an intersection of the possible events of each endpoint.

---

> > ### Author Rebuttal · Reviewer_C4hX · 2026-04-03
> >
> > Thank you for your thorough response; my questions and concerns have been well-addressed and hence, I am pleased to maintain my score.
> >
> > As a further note, I believe a remark discussing weakening of the concept-factorization assumption that follows the theorem (as  done in (1) above) would further strengthen the paper.

---

> > > ### Author Response · Authors · 2026-04-04
> > >
> > > Thank you for your support and thank you for the suggestion. We will add more commentary on the concept-factorization assumption in the revised version of the paper.

---

### Decision · Program_Chairs · 2026-04-30

**Decision:**

Accept (regular)

**Comment:**

Softmax outputs should be viewed as probability distributions endowed with an information-geometric structure induced by the KL divergence, rather than as Euclidean vectors. This perspective induces a Bregman, dually flat geometry on the logit space, where representations correspond to the natural parameters of an exponential family and admit a dual coordinate system via convex conjugacy. Within this framework, one can distinguish two semantically different modes of interpolation. In particular, directly adding a probe vector in the primal space leads to off-target drift, whereas the proposed dual steering approach more effectively preserves off-target distributions under a concept-factorizability assumption. These findings are supported both theoretically and empirically.



The paper addresses an interesting and timely problem and proposes a principled and elegant approach with clear insights that is also practically relevant. The theoretical results are sound and valid, although they rely on some strong assumptions that may not always hold in practice. The empirical validation is thorough and well executed, effectively demonstrating the benefits of the method. The paper is generally well written, with a clear motivation and only a few issues that require adjustment. Overall, the proposed methodology appears novel and significant.


Some concerns have been raised regarding the assumptions of the method, as well as the consideration of only the last layer and the absence of comparisons with alternative approaches. The scalability of the method has also been questioned, and the fact that the experimental evaluation is conducted in relatively controlled settings and could be more extensive. Some relevant related work appears to be missing, while the limitations should be discussed more extensively and the actual algorithm should be included.


The authors have addressed the reviewers’ concerns to a satisfactory extent, which was acknowledged and appreciated. The consensus is that the paper is a significant contribution, even though some parts need to be updated. For this reason, I recommend acceptance. That said, the authors should carefully incorporate the feedback and revise the paper accordingly.